# Glacial lake outburst floods threaten millions globally

Caroline Taylor[1], Tom R. Robinson ![ORCID][2] ✉, Stuart Dunning[1], J. Rachel Carr[1] & Matthew Westoby ![ORCID][3]

Glacial lake outburst floods (GLOFs) represent a major hazard and can result in significant loss of life. Globally, since 1990, the number and size of glacial lakes has grown rapidly along with downstream population, while socio-economic vulnerability has decreased. Nevertheless, contemporary exposure and vulnerability to GLOFs at the global scale has never been quantified. Here we show that 15 million people globally are exposed to impacts from potential GLOFs. Populations in High Mountains Asia (HMA) are the most exposed and on average live closest to glacial lakes with ~1 million people living within 10 km of a glacial lake. More than half of the globally exposed population are found in just four countries: India, Pakistan, Peru, and China. While HMA has the highest potential for GLOF impacts, we highlight the Andes as a region of concern, with similar potential for GLOF impacts to HMA but comparatively few published research studies.

Glaciers are particularly sensitive to changes in climate[1–3] and are highly visible indicators of climate warming[3–5]. Over the last three decades there have been substantial decreases in global glacier mass, with ice losses between 2006 and 2016 estimated at $-332 \pm 144$ Gt y$^{-1}$[6,7]. This decline is likely to persist through the 21st century as most glaciers are out of balance with present climate; ~$36 \pm 8\%$ of current mass loss is a 'lagged response' to past climate forcing[8]. In many areas, over-deepenings in former glacier beds are uncovered during the course of glacier retreat, which allows melt water to collect as glacial lakes[9–11]. Glacial lakes can also form via the growth and coalescence of supra-glacial ponds on debris-covered glaciers[12,13], and in other ice-marginal settings[14,15]. The formation of glacial lakes can trigger positive feed-backs, whereby lakes promote further ice loss through calving and subaqueous melting, causing additional melt and retreat, and further lake expansion[16–18].

Importantly, these lakes can represent a substantial hazard in the form of glacial lake outburst floods (GLOFs). GLOF triggering is complex, with dam breach initiation caused by mass movement-induced impulse waves[19,20], lake overfilling due to pluvial, nival and glacial runoff[21], and moraine- or ice dam degradation being variably important dependent on setting[22,23]. Consequently, the probability of a lake releasing a GLOF is difficult to accurately quantify without detailed and localised studies.

GLOFs can be highly destructive and can arrive with little prior warning, causing significant damage to property, infrastructure, and agricultural land, and resulting in extensive loss of life. However, the impact varies significantly across the globe; in the last 70 years, several thousand people have been killed by GLOFs in the Cordillera Blanca alone[24,25], most from a small number of events[26,27], while only 393 deaths in the European Alps can be directly linked to GLOF activity over the last 1000 years[28]. The continued ice loss and expansion of glacial lakes due to climate change therefore represents a globally important natural hazard that requires urgent attention if future loss of life from GLOF is to be minimised[29,30] and the UN's Sustainable Development Goals (particularly Goal 11—Disaster Risk Reduction) are to be met.

Since 1990, the number, area, and volume of glacial lakes globally has grown rapidly, increasing by 53%, 51%, and 48% respectively[30]. Concurrent with the rapid growth of glacial lakes, many catchments downstream have experienced rapid and large increases in population, infrastructure and hydroelectric power (HEP) schemes, while agriculture has intensified[31–35]. However, the socio-economic vulnerability

[1]School of Geography, Politics and Sociology, Newcastle University, Newcastle upon Tyne, UK. [2]School of Earth & Environment, University of Canterbury, Christchurch, New Zealand. [3]Department of Geography and Environmental Sciences, Northumbria University, Newcastle upon Tyne, UK. ✉ e-mail: thomas.robinson@canterbury.ac.nz

to climate-related hazards is thought to have decreased[36], although this decrease is spatially heterogenous and it remains unclear if this heterogeneity is sufficient to offset potential increases in hazard and exposure. Contemporaneous changes in lake conditions and downstream damage potential (i.e., the combination of exposure—the proximity of populations to a potential outburst—and vulnerability—the exposed populations likelihood to be impacted by the GLOF) are all critical components of GLOF danger[10,31,37,38]. However, how the recent observed changes in each combine to produce contemporary global GLOF danger remains unclear[29.] While regional scale GLOF risk assessments have been undertaken[39,40], to our knowledge, no global scale study has been attempted that considers not just the physical lake conditions, but also societal exposure and vulnerability that directly influence GLOF danger[41].

Here we combine the most up-to-date lake condition, exposure, and vulnerability data available to quantify and rank contemporary (2020) damage potential from GLOFs at a global scale, adding to similar recent approaches for hydrometeorological floods[42,43]. We analyse the spatial distribution of population exposure to determine where populations are in relation to glacial lakes, using necessarily simple estimates of potential GLOF runout paths (50 km runout, with potentially affected populations located within 1 km of a river course), therefore identifying potential GLOF danger hotspots and thus higher priority zones for mitigation and further, local-scale research. While this study captures lake conditions and damage potential as they were in 2020, the methods presented provide a framework to capture changing GLOF danger through time.

## Results

### Lake conditions
As of 2020, regional normalised GLOF lake conditions, represented in terms of the total number and area of glacial lakes, were highest in the Pacific Northwest (PNW; 1.000), and lowest in the European Alps (0.041) (Fig. S1). There was high variability between nations, with individual GLOF lake conditions highest in Greenland and Canada (1.000 and 0.685 respectively) and lowest in Ecuador (0.001). Excluding Uzbekistan (no glacial lakes, normalised hazard score of zero) the largest range in intra-regional GLOF lake condition scores were seen in High Mountain Asia (HMA), ranging from a high score in China (0.319) to a low score in Mongolia (0.006). Generally, normalised national GLOF lake condition scores in HMA are below 0.100, with the exception of China.

### Exposure
In total, 90 million people across 30 countries live in 1089 basins containing glacial lakes (Fig. 1a). Our analysis indicates that of these, 15 million (16.6%) live within 50 km of a glacial lake and 1 km of potential GLOF runout tracks (Fig. 1a). We find that 62% (~9.3 million) of the globally exposed population are located in the HMA region. Globally, the proportion of exposed population varies significantly between countries; India and Pakistan contain the highest number of exposed people (~3 million and ~2 million people respectively, or one-third of the global total combined) while Iceland contains the least (260 people) (Fig. 1b). Just four highly populous countries account for >50% of the globally exposed population: India, Pakistan, Peru, and China (Fig. 2a). As a result, regionally HMA has the highest normalised exposure score (1.000) while the High Arctic and Outlying Countries score the lowest (0.019). India and Pakistan are the highest individually scoring nations (1.000 and 0.701), and Sweden is the lowest (0.001).

Generally, the population exposed to GLOFs increases with distance from a glacial lake, with almost half (48%) of exposed populations globally located between 20 km and 35 km downstream of lakes (Fig. 2). Only 2% (300,000) of the global population exposed to GLOFs live within 5 km of one or more glacial lakes (Fig. 2), with the majority of these (66%; 198,000) found in HMA (Fig. 2). Populations in HMA live,

on average, closer to glacial lakes than anywhere else, with ~1 million people living within 10 km downstream of a glacial lake, where any early warning time is likely to be low, and, uncertainty in GLOF magnitude high. In contrast, populations across the PNW and High Arctic and Outlying Countries are generally situated further than 35 km downstream from glacial lakes (Fig. 2). Analysis of exposure at the national scale reveals considerable sub-regional variability, with populations in Pakistan living closest to glacial lakes, (0.8 million within the first 15 km (Fig. S2)), while settled populations in Kyrgyzstan are living at least 35 km downstream (Fig. S2).

### Vulnerability
The three indices which were used to calculate vulnerability (CPI, HDI, and SVI) showed marked variation between and within regions (Fig. S3). Generally, the Andes and HMA have the highest levels of corruption and social vulnerability and lowest levels of human development, while the contrary is true for the European Alps, PNW and High Arctic and Outlying Countries. However, regional summaries mask some substantial national variations; although the Andes region has an average corruption score of 51, country level scores vary from high corruption in Bolivia (33) to lower corruption in Ecuador (88). Similarly, the average human development score in HMA (0.671) masks a range of scores, from a low of 0.511 (Afghanistan) to a high of 0.825 (Kazakhstan). Nonetheless, HMA is identified as the most vulnerable region to GLOF in 2020 (0.768) and the PNW the least (0.336). Overall, Afghanistan and Pakistan are the most vulnerable nations (0.919 and 0.837 respectively) while Switzerland and New Zealand are the least (0.194 and 0.186 respectively).

### GLOF Danger
The combined normalised scores of GLOF lake conditions, exposure and vulnerability reveal HMA to have the highest GLOF danger as of 2020 (0.313), with a total of 9.3 million people exposed to 2211 lakes covering an area 1256.09 km$^2$. Comparatively, the High Arctic and Outlying Countries have the lowest GLOF danger (0.032) with <200,000 people exposed, albeit to a similarly high number and area of glacial lakes (1862 lakes covering an area 1166.09 km$^2$) (Fig. 3). As with the individual components, there is substantial sub-regional variation in GLOF danger itself (Fig. 3). China and Pakistan have the highest danger globally (0.863 and 0.751 respectively). Pakistan has near double the exposed population of China (2.1 million and 1.1 million respectively) and is significantly more vulnerable (0.837 compared to 0.683 in China). However, with more numerous lakes, and of larger area (1109 lakes covering 1094.44 km$^2$) the GLOF lake condition score in China is large enough to more than offset these differences. Both Greenland and Uzbekistan have a danger score of zero because as of 2020 they have, respectively, no exposed population and no glacial lakes.

When all 1089 glacial basins are ranked from highest to lowest risk (Fig. 3), the top three are found in Pakistan (Khyber Pakhtunkhwa basin), Peru (Santa basin), and Bolivia (Beni basin) (Fig. S6, Table S1) containing, respectively, 1.2 million, 0.9 million and 0.1 million people who could be exposed to GLOF impacts. Interestingly, Canada and USA contain just 3 basins in the top 50 globally (Table S1) as well as the lowest ranking basin (Tyers basin, Canada), where exposure is negligible as potential GLOF runout tracks are largely unpopulated. However, Canada and USA have relatively high GLOF danger scores (0.321 and 0.059) ranking 4th and 6th respectively, mainly because they host a large number of basins with generally high GLOF lake condition scores, highlighting the importance of spatial scale in these analyses.

## Discussion
With an increase in interest surrounding GLOFs over the last few decades, a clear geographical disparity has emerged between where GLOFs are occurring and the hotspots of research[44,45]. Between 1990

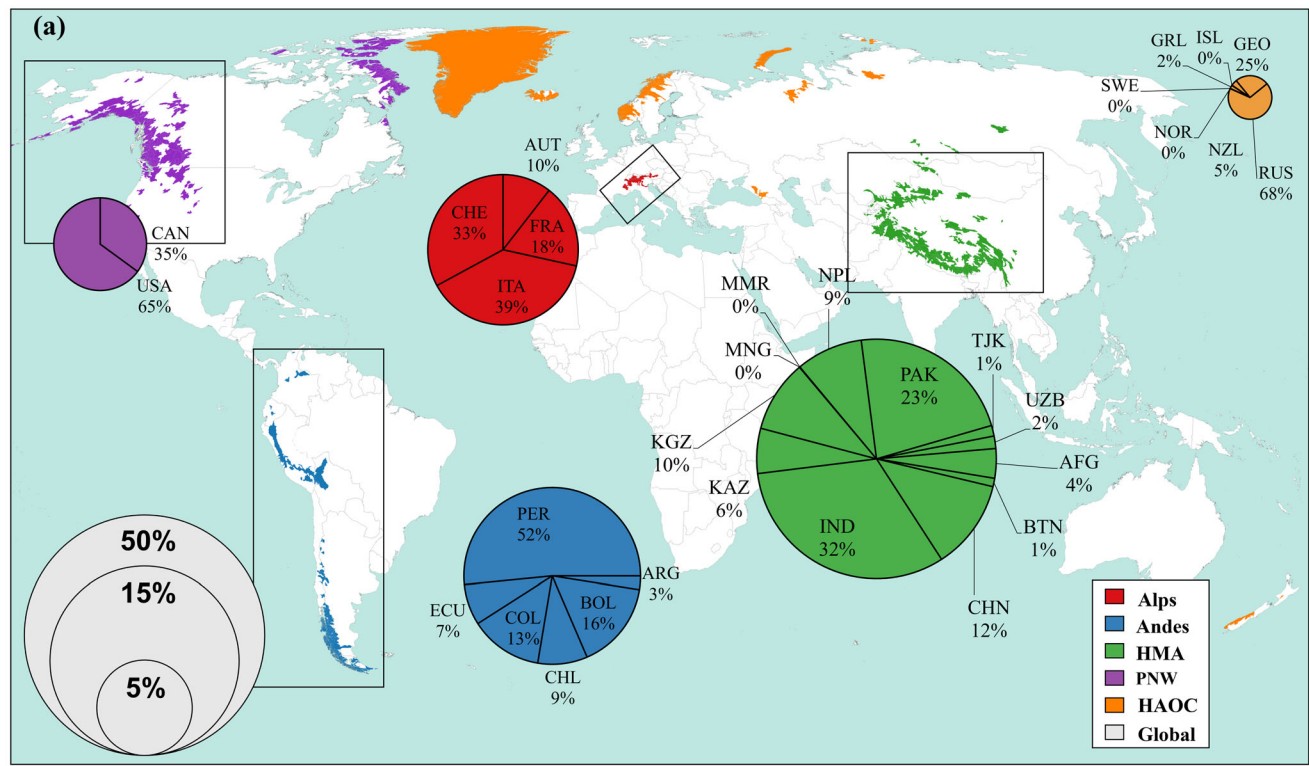

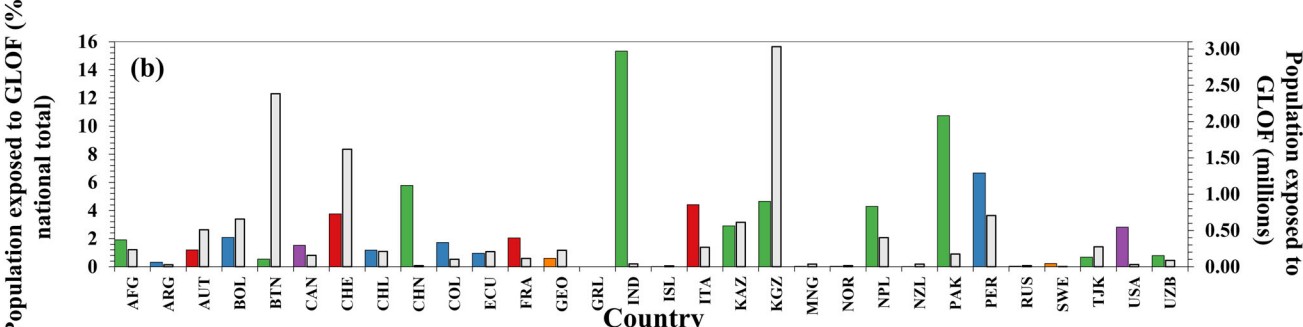

**Fig. 1 | Global distribution of GLOF exposure. a** Global distribution of glacial basins, colour-coded according to mountain range, with 'High Arctic and Outlying Countries' (HAOC) representing all basins outside of the four main ranges in this study (Alps, Andes, High Mountains Asia (HMA) and Pacific North West (PNW)). Pie charts show the proportion of exposed population as individual country contributions to the mountain range total, with pie charts sized according to percentage contribution to the 2020 global total. **b** Grey bars show exposed population as a percentage of the national total (left axis). Coloured bars show the total exposed population per country (right axis).

and 2015, Iceland, the North American Cordillera and Hindu-Kush-Karakoram were the most prominent GLOF research hotspots with 180, 144, and 142 published research items, respectively[44] (Fig. S3). Since 2015, however, the Himalayas have emerged as the primary research focus, accounting for 36% of the studies undertaken between 2017 and 2021[45]. As such, these 'hotspot' regions are often cited as having the highest GLOF danger. While true in part, our results also indicate that as of 2020, the potential for large GLOF impacts is also high across the Andes (Fig. 3), and as a nation, danger in Peru is third highest globally (Fig. 3b).

Over the last two decades, glaciers across the Andes have undergone rapid deglaciation in response to climate changes[46,47] leading to the growth of many large glacial lakes and consequently a growth in overall GLOF lake conditions (Fig. 4); the number of glacial lakes across the region increased by 93% compared to just 37% in HMA across the period. Concurrent with this increase, populations living in close proximity to glacial lakes have grown (Fig. 4), increasing the overall exposure to GLOF (Fig. 2); since 1941 the population in Huaraz, Peru alone has increased by >100,000[48]. At the same time, regional

vulnerability remains high as a result of deep-rooted corruption and poor standards of living (Fig. S4). Comparative to other regions, the number of GLOF research items across the Andes are few; less than 8% of the research conducted between 1979 and 2021 were undertaken in this region (<100 items)[44,45] (Fig. S3). We suggest this data sparsity across the Andes is perhaps preventing meaningful assessments of actual GLOF risk in the region and urgently requires attention, particularly given the second- and third-most dangerous basins are found in this region, and the region as a whole is ranked second for GLOF danger globally (Fig S6).

GLOFs can have exceptional discharges and runout distances, reaching >120 km downstream[49,50]. Here, we have considered anyone living within 1 km of likely GLOF runout tracks up to a maximum distance of 50 km from the glacial lake to be at risk of either direct (e.g., death or injury) or indirect (e.g., loss of land, damaged infrastructure) impacts. However, peak discharge attenuates rapidly from the flood source[31], meaning that impacts are generally greatest with increasing proximity to a glacial lake. We show populations in HMA (Fig. 2), and particularly those in Pakistan (Fig. S2), are living closest to glacial lakes.

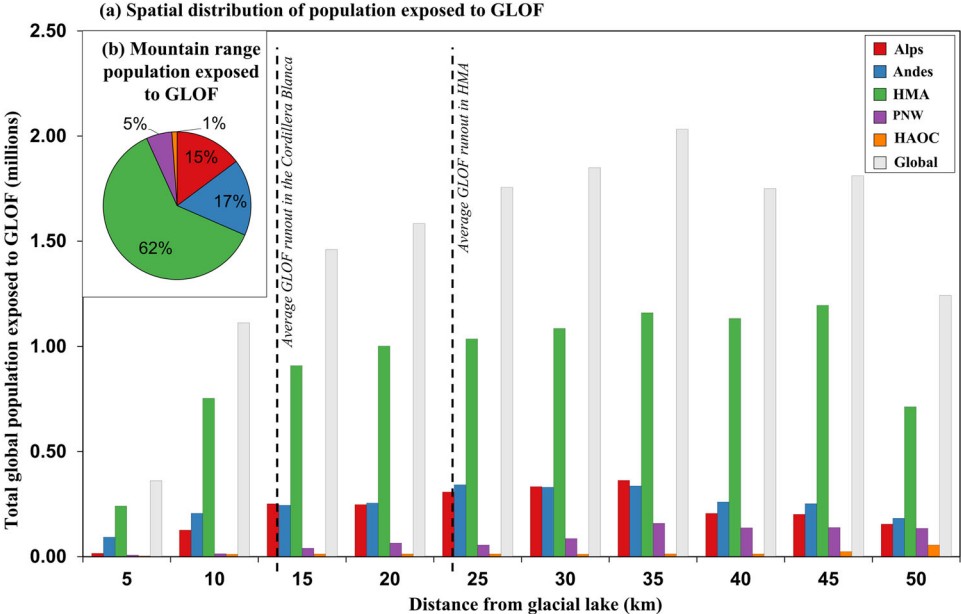

**Fig. 2 | Global spatial distribution of exposure. a** Spatial distribution of exposure within GLOF runout tracks up to 50 km from a glacial lake, at 5 km intervals at the global and mountain range scale. **b** Total contribution of mountain range to the global total exposed population. Countries are coloured according to mountain range. *HMA* High Mountains Asia, *PNW* Pacific North West, *HAOC* High Arctic and Outlying Countries.

With the expansion of agriculture, development of new HEP sites (and with increasing proximity to glacial lakes), and growth of the tourism sector expected to increase in this region over the next few decades, it follows that exposure is only likely to increase as people move to higher elevations to support the aforementioned development, as has been observed in other mountain regions globally[24,51–53]. The characteristic rapid onset and high discharge of GLOFs means there is often insufficient time to effectively warn downstream populations and for effective action to be taken, particularly for populations located within ~10–15 km of the source lake[54,55]. Improvements are urgently needed to Early Warning Systems (EWS) alongside evacuation drills, plus other forms of community outreach that are sympathetic to potential social and cultural barriers, to enable more rapid warnings and emergency action in these highly exposed areas. Across HMA resources for mitigation are often limited[28], and residents' lack of awareness, or lack of means to affect change, inhibits their ability to prepare for, and recover from, potential GLOF disasters sourced from remote glacial lakes[56]. Thus, analysing the spatial distribution of exposure as presented here not only highlights where advances are needed (e.g., EWS) but could also allow for more effective mitigation strategies (e.g., land zoning, education) to be implemented. Similar to our findings for the Andes, while Pakistan is a hotspot of GLOF danger, there is a comparative lack of published research focusing on this country and despite large-scale investment (>US$30 million) in GLOF vulnerability projects from the United Nations Development Programme[57]. We suggest the area should be targeted for more detailed research.

Glaciers are exhibiting negative mass balance in nearly all glaciated regions of the world[5], and over the past three decades the number, area and volume of glacial lakes have increased rapidly[30]. Our data show that countries with the largest, or most numerous, glacial lakes do not always possess a high GLOF danger. Instead, our results show that it is the exposed population that greatly elevates the potential impact of GLOFs globally (Fig. 4), particularly across HMA and the Andes (Fig. 1). For instance, Greenland has the highest number and area of glacial lakes of any nation in this study, thus has the highest hazard score (1.000) yet no people reside along likely GLOF runout tracks giving it a danger score of zero. Documenting changes in glacial lakes and highlighting areas where GLOF lake conditions may be

increasing, while valuable, does not therefore provide an accurate indication in terms of danger trajectories, since contemporaneous changes in population exposure may more than offset changes in lake conditions (Fig. 4). Furthermore, our data begins to highlight the degree to which natural disasters impact people; two outburst events affecting the same number of people with the same material impact (e.g., a footbridge or road washed away) can have fundamentally different consequences depending on the social, political, cultural and economic context of the country, or even catchment, in which they occur[24,28,58–60]. This highlights the crucial role of exposure and vulnerability in determining the impact of GLOFs. While hazard assessments dominate GLOF studies[61], exposure and vulnerability assessments remain relatively unexplored topics that urgently need addressing, particularly in developing countries[62] where GLOF danger is generally highest (Fig. 3).

How GLOF danger might change in the future remains subject to debate. As glaciers continue to recede existing glacial lakes will expand, and many new lakes will form[39], altering the spatial pattern of GLOF lake conditions[63]. At the same time, we will see spatiotemporal changes in populations and their vulnerability as people, goods and services migrate in response to various socioeconomic drivers, and development related to the growth of tourism, HEP and agriculture continues to expand into higher elevations closer to glacial lakes and other forms of natural hazard[52]. We have shown the most dangerous basins, mainly found across HMA and the Andes do not always host the most, or the largest, glacial lakes, and rather it is the high number of people and the reduced capacity of those people to cope with disaster that plays a key role in determining overall GLOF danger (Fig. 3). This finding highlights the need for a more holistic approach to GLOF risk assessment, where each component of hazard, exposure, and vulnerability are accounted for. We highlight the value of global-scale spatial danger analysis (Fig. 3) and envisage our findings to be the starting point for more targeted risk assessments at the national- and basin-scale. Our findings are important, as they not only identify countries and basins that rank highly in terms of GLOF danger, which can allow for more targeted GLOF risk management, but also regions where more research is urgently needed to understand risk at a fundamental level. In particular, we highlight the Andes as an under-

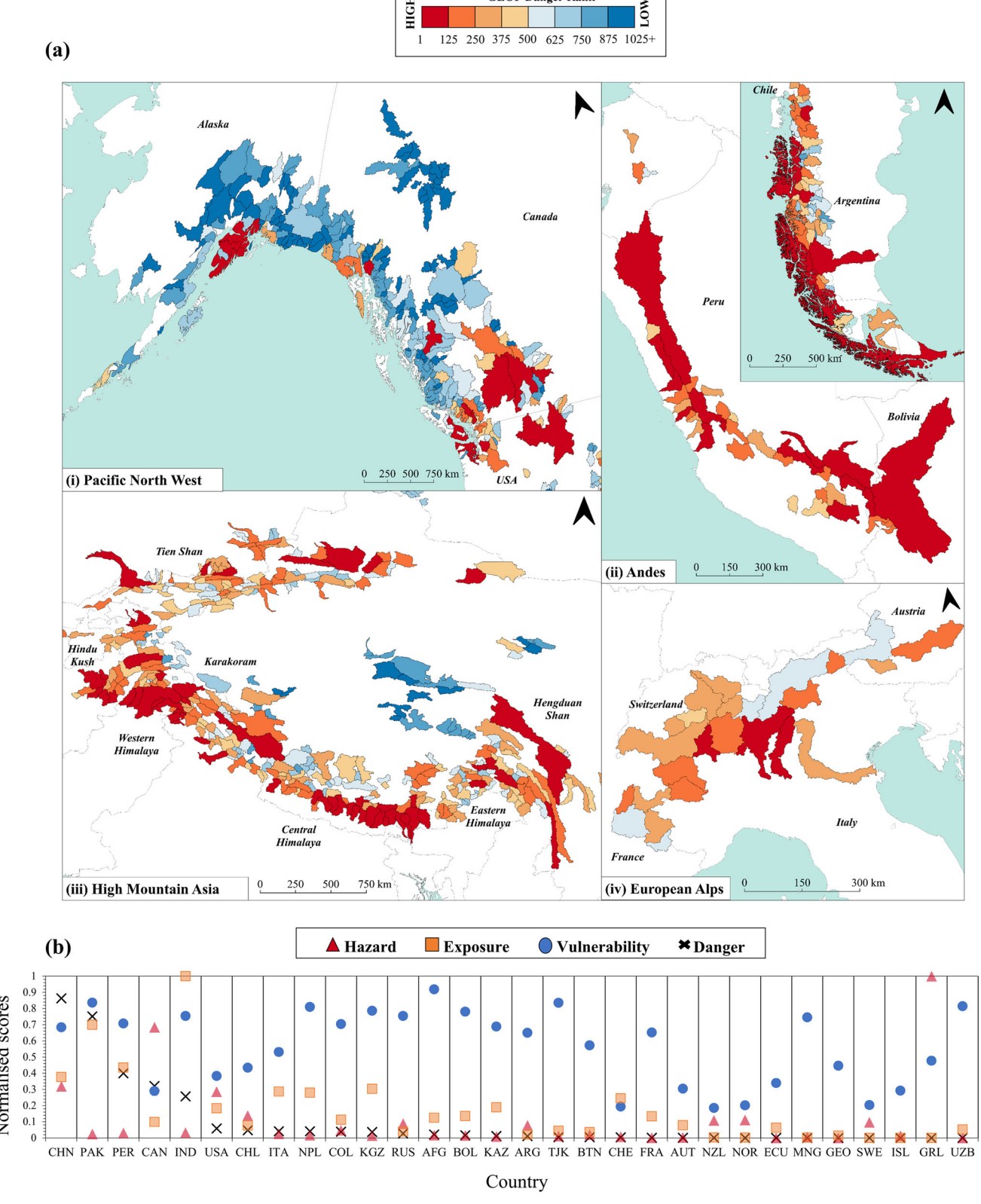

**Fig. 3 | Global GLOF danger. a** Spatial distribution of GLOF danger at basin scale from high (red) to low (blue) risk. **b** Final normalised scores of GLOF lake conditions ('hazard'), exposure, vulnerability, and danger for each country, ordered from highest danger score (left) to lowest (right).

studied hotspot of GLOF danger and suggest that the region is targeted for more detailed study. While we show the global picture of contemporary GLOF danger, it remains unclear how this danger is changing temporally and whether such changes are being driven by

changes in lake conditions, damage potential, or some combination. Thus work is required to evaluate temporal changes in lake conditions, exposure and vulnerability in order to determine the relative roles in each for GLOF danger.

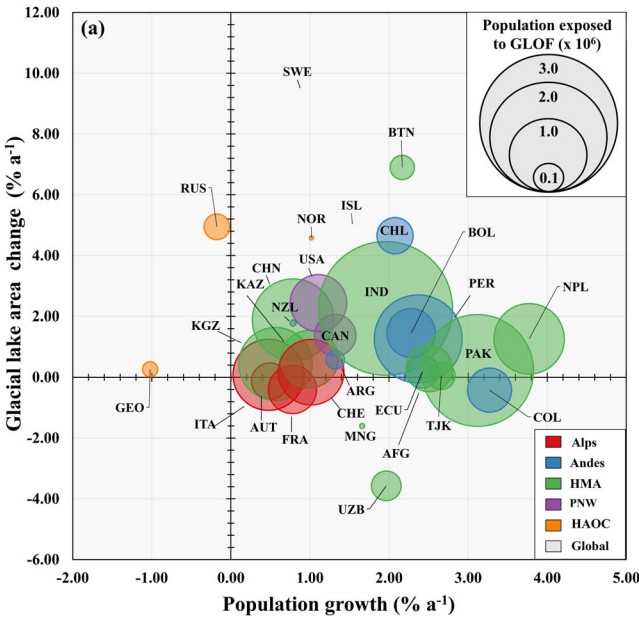
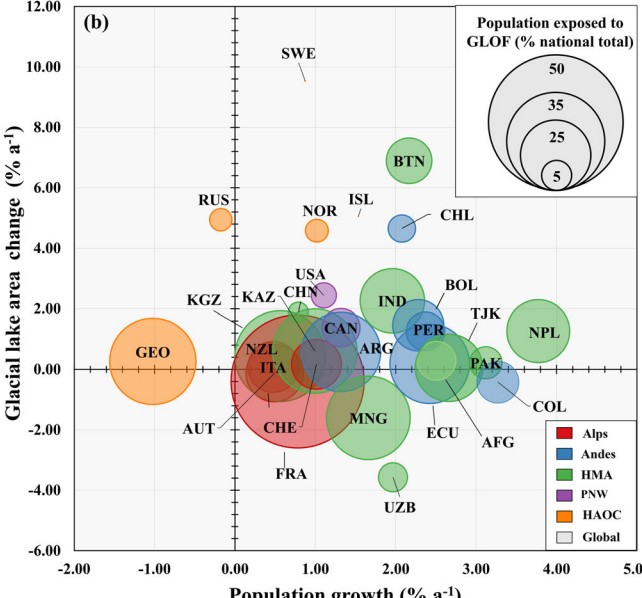

**Fig. 4 | Rate of change in glacial lake area and total population.** Rates of population change between 1990 and 2020 and glacial lake area change between 1990 and 2018 as **a** absolute population exposed to GLOFs and **b** percentage of national population exposed to GLOFs in each country. Countries are colour coded according to Mountain Range. While our study only considers contemporary GLOF danger, this highlights that variable changes in population and lake conditions may lead to very different danger scores in the near future. *HMA* High Mountains Asia, *PNW* Pacific North West, *HAOC* High Arctic and Outlying Countries.

## Methods

### GLOF lake conditions

Within natural hazard research, hazard is a critical component of risk and is defined as a function of the probability and intensity of an event, i.e., the likelihood that an event will occur from a given site based on intrinsic properties and dynamic characteristics of that site combined with the overall magnitude of the event[52]. Thus, the probability of a GLOF occurring at a given point in time is dependent on specific local conditions, including, but not limited to; potential topographic triggers (ice/rock/snow avalanche etc), lake-dam geometries, and lake area/volume etc[10,29,33,39]. Further, the likelihood of lake failure will almost certainly vary temporally. Attempts to quantify the probability of GLOFs have been undertaken at regional-scale using simple proxies for the likelihood of landslide and/or ice avalanches into lakes[39,53]. However, to be applied globally, these approaches require globally consistent, high-resolution DEMs, which are known to suffer from considerable artefact issues in high mountain regions where GLOFs originate[64]. Therefore, quantifying the probability of failure is inherently difficult at a global scale. Thus, here, we take a consequence-based approach and focus on quantifying only the intensity of a potential GLOF. We do this by using the total lake area as a proxy for intensity, where larger lakes have the potential to produce larger, more intense GLOFs. Previous regional-scale work[39] has sought to use lake volume as a proxy for GLOF intensity by applying simple area-volume relationships to convert mapped lake area to assumed lake volume. However, for our study any area-volume relationship would need to be globally consistent, scaling all lake hazard values consistently, and thus we prefer to use simply mapped lake area instead.

As such, we treat the probability of failure as unknown and instead focus on quantifying the impacts or effects on the potentially affected population. Consequently, we prefer the term GLOF lake conditions to GLOF hazard, with our scoring system instead highlighting conditions that may yield more intense GLOFs should a failure occur. When combined with damage potential, the basins with the largest potential impacts can then be targeted for more detailed local studies to ascertain the probability of a GLOF occurring in the first place, allowing those with the highest potential losses to be prioritised for more local

studies. While probability is a key element of risk, we note that basins with high potential impact would still be considered comparatively high risk if the probability of failure was found to be low, while basins with low potential impacts would only suffer marginal increases in comparative risk if the probability of failure was found to be high.

We use the Level 4 Global Water Resource Zones shapefiles[65] and the most recently available global inventory of glacial lakes[30] to identify 1089 basins containing glacial lakes. We note that these Water Resource Zones do not represent true river catchments, instead showing regions that contain several associated rivers flowing into a lake or ocean, with Level 4 representing rivers that have no tributaries larger than 100 km². This can cause strange effects, particularly in large coastal or plains areas, such as in Chilean Patagonia (Fig. 3a). However, to our knowledge there is no suitable global dataset of river catchments, and regional and national datasets are too inconsistently derived for our globally-focussed study. We group basins into four main mountain ranges; European Alps, Andes, High Mountain Asia (HMA) and Pacific Northwest (PNW), with the remaining 131 (12%) basins outside of these ranges referred to as 'High Arctic and Outlying Countries'. We then extract the raw number and area of glacial lakes per basin/country/region to act as proxies for potential GLOF intensity, before performing a linear transformation function to produce a normalised value for each indicator (Eq. 1);

$$y_{N/A} = \frac{(X)}{(Max)} \qquad (1)$$

where $x$ is the absolute number/area of glacial lakes per basin/country/region, Max is the maximum number/area of glacial lakes found out of all basins/countries/regions, and $y$ is the normalised value of glacial lake number/area per basin/country/region. Individual normalized values of glacial lake number ($y_N$) and area ($y_A$) are then multiplied to produce a singular score between 0 and 1, with higher values relating to lake conditions with the potential for more intense GLOFs.

Finally, we consider the potential downstream spatial extent of GLOFs by considering the expected reach. Runout distances of GLOFs primarily vary as a function of outburst volume and stream

gradient, as well as other factors such as bed roughness, sediment concentration etc[66]. Thus, defining a runout distance from which to assess exposed population on a global scale is difficult. Previous research[29] set a runout cut-off distance of 50 km, to facilitate a standardized comparison between glacial lakes. Their 50 km threshold is consistent with a number of observed runout distances of past GLOFs, such as at Dig Tsho in 1985[67], Chilleon Valley in 2015[68] and Chorabari in 2014[69]. Comparisons of likely GLOF discharges with that of meteorological floods[70] suggest the majority (50%) of likely GLOFs that exceed the 100-year meteorological flood discharge do so to only ~20 km downstream, with 1% theoretically reaching >85 km[31]. However, with lake sizes increasing due to climate change, runout of future GLOFs may well exceed that of those previously observed due to the larger volume of water potentially involved. Nevertheless, although we recognise runout distances vary considerably, with some GLOF events showing runout length >200 km[50], considering such distances at a global scale could lead to large overestimations of downstream impacts in many locations[29]. So, following the approach of Dubey & Goyal (ref. [29]) we use a cut off distance of 50 km, which should encapsulate the majority of runouts globally and provide a conservative estimate of potential GLOF reach accounting for potentially longer runout GLOFs in the future, while avoiding large overestimations by using observed but rare extreme runout distances. Using a 50 km cut off distance also accounts for issues arising from our use of Water Resource Zones, as only the region within 50 km of a glacial lake are assessed. In coastal and plains areas where the Water Resource Zones can incorporate areas without glacial lakes upstream, this ensures that only the area and population downstream and in proximity to glacial lakes are included in our calculations.

### Exposure

GLOF runout pathways tend to follow river channels[28,71], so impact increases with proximity to the channel[72]. Thus, similar to previous approaches[73] we further constrain our potential GLOF footprint to estimate exposed populations by applying a 1 km buffer either side of any main river channel[65] with a glacial lake in its upper reaches, up to a distance of 50 km (Fig. S5f). We used the 2020 Gridded Population of the World version 4 (GPWv4)[74] to sum the population count per 1 km$^2$ cell within this buffer, obtaining exposed population (Fig. S5h). We recognise that a 1 km buffer is a crude estimate for identifying potential GLOF impact zones; exposed population is likely overestimated in the upper reaches where steeper elevations and narrow river valleys likely mean populations within even 100 m of a river channel may in fact be far above the impacted zone, while in the lower reaches where valleys are flatter and wider, exposed population is likely underestimated. However, as the overall impact of a GLOF wanes with distance from the river channel[72,73], and given the resolution of the population data used[74], at a global scale a 1 km buffer will provide a conservative but consistent estimate of the potentially exposed population. These areas of concern can then be targeted for further, more detailed analysis using more complex GLOF runout modelling and higher resolution population data to refine our initial estimates. We use a linear transformation function to produce a normalised value of exposure for each basin (Eq. 2);

$$E = \frac{(P)}{(\text{Max})} \tag{2}$$

Where $E$ is the normalised exposure score, $P$ is the total exposed population per basin/country/region, and Max refers to the maximum exposed population per basin/country/region respectively. To add further granularity, we split the 50 km buffer into 5 km intervals and summed the population within these intervals, to determine how population is distributed along these likely GLOF runout tracks.

### Vulnerability

Many factors influence human vulnerability to natural hazards[75–77], and yet, due in part to the absence of sufficient data, few studies have considered the temporal trend in vulnerability[78]. Since the implementation of the Millennium Development Goals and the succeeding Sustainable Development Goals, there has been a vast improvement in the amount, and quality, of vulnerability data available. Here we combine qualitative information obtained from the Corruption Perception Index (CPI) at national-scale and Human Development Index (HDI) at sub-national level (first internal administrative level, e.g., state or province) with a national-scale Social Vulnerability Index (SVI) to provide a proxy for GLOF vulnerability. At a global scale, corruption and human development are indicative of population fragility[79–81] with higher levels of corruption and lower levels of development individually associated with larger impacts. The CPI scores and ranks countries/territories based on how corrupt a country's public sector is perceived to be by experts and business executives. It is a composite index comprised through 13 data sources and is the most widely used indicator of corruption worldwide. The HDI is a summary measure of three key dimensions of human development: health, education, and standard of living[82], and is comprised of normalised indices of: life expectancy, expected years of schooling, mean years of school and Gross National Income (GNI) per capita. Both the CPI and HDI have been successfully used in previous natural hazard risk assessments[51,83].

While both the CPI and HDI provide a useful metric for assessing the development of a country/territory[83], they do not reflect on many factors that influence social vulnerability[75]. Thus, to assess the coping capacity of downstream communities and the ability of the affected nation to effectively respond to the event, a SVI was also calculated. Drawing upon an existing flood vulnerability assessment[84], the SVI used in this study initially analysed 9 indicators (Table S2) that either reduce or enhance a population's and nation's capacity to cope with a GLOF disaster. To avoid double counting, we performed a correlation study (matric-plot and correlation-matrix) to ensure variables were independent from other indicators as well as those used to calculate the HDI and CPI. To keep the sample size valid, preference was given to variables with the lowest number of missing variables. As a result, four variables from the SVI were not included when calculating the final vulnerability score; percentage of safe drinking water and percentage of good sanitation as well as percentage illiterate population and percentage unemployment. The former two were highlighted both for double counting and lack of datapoints, and the latter two for double counting with data used to calculate the HDI. Consequently, the final SVI score was based on 5 unique indicators (Eq. 3);

$$SVI = \frac{\left(\frac{\text{reducing indicators}}{\text{enhancing indicators}}\right)}{5} \tag{3}$$

We acknowledge that proxy values of vulnerability at national- and sub-national-scale will hide more granular variations within countries. However, vulnerability data at finer resolution is largely absent globally and therefore we argue our approach provides the highest globally consistent resolution currently available for a global-scale study. Furthermore, we note that while the vulnerability of the immediately exposed population is critical to understanding the eventual impacts from a disaster, the capacity of the country as a whole to adequately respond to the disaster is also an important factor. As such, our vulnerability indicators attempt to capture both the physical vulnerability of the directly exposed populations, and the capacity of the country/region as a whole to cope with the event.

Finally, we note that the relative importance of each indicator on social vulnerability will change with location, with studies often assigning weights using an analytic hierarchy process and expert knowledge to fit the specific context of the study[84]. Given the global scale of this study, an 'equal weighting' approach was selected with the

understanding that the outputs should be taken as a baseline value, and exact values per country may vary. In this study, all three indicators (HDI, CPI, and SVI) are normalised and combined with equal weighting (Eq. 3) to produce a single proxy for vulnerability (Eq. 4). Final values range between 0 and 1, where 1 equates to the highest vulnerability. No scores of absolute 0 were recorded.

$$\text{Vulnerability} = 1 - [\text{HDI} \times (1 - \text{CPI}) \times \text{SVI}] \qquad (4)$$

### GLOF danger

The normalised results of all three parameters (GLOF lake conditions, exposure, and vulnerability) were then combined to produce a semi-quantitative metric for GLOF danger (Eq. 5). Here, we prefer the term GLOF danger over GLOF risk due to the lack of probability in our GLOF lake conditions score. Basins were then ranked from highest (1) to lowest (1089) danger to identify hotspots of GLOF danger.

$$\text{GLOF danger} = \left[ \text{Lake Conditions} \times \text{Exposure} \times \text{Vulnerability} \right] \qquad (5)$$

## Data availability

All the data used in the study are available from open-source repositories. Glacial lake data files spanning 1990-2018 are available from https://nsidc.org/data/HMA_GLI/versions/1. Population data are available at https://doi.org/10.7927/H4X63JVC. National corruption scores are available from Transparency International at http://www.transparency.org/en/cpi/2019. Sub-national human development scores are available from the United Nations Development Programme (UNDP) at https://hdi.globaldatalab.org/areadata/. Data for the indices used to derive SVI are available from the World Bank Open Data at World Bank Open Data | Data. The Global Water Resource Zones are available from https://doi.org/10.6084/m9.figshare.8044184.v6.

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

## Acknowledgements

This work was supported by the Natural Environment Research Council (NERC)- funded ONE Planet Doctoral Training Partnership [NE/S007512/1]. We thank Professor Andy Large and Dr Jon Telling for their critical reading and suggestions for the manuscript.

## Author contributions

C.T., T.R., R.C., and S.D. devised the study. C.T. undertook the computational studies and data analysis. T.R., R.C., S.D., and M.W. supervised the work. All authors wrote and edited the manuscript.

## Competing interests

The authors declare no competing interests.
