## [Peer Review File · Nature Communications]

REVIEWER COMMENTS

Reviewer #2 (Remarks to the Author):

NOTEWORTHY RESULTS

The manuscript addresses the assessment of current risks posed by glacial lakes to downstream population at global scale. Risk posed by glacial lakes is a hot topic in earth- and geosciences, but also climate change impact research: Since several years, abundant studies are published on related topics at local to regional (continental scale) on glacial lakes and related hazards. This is the first study, aiming at assessing the risk of glacial lakes outburst floods (GLOFs) at global scale, and thereby not only focusing on the physical aspects of risk (i.e. hazard), but considering also socio-economic drivers of risk, by determining exposure and vulnerability of the population living downstream of such glacial lakes. Results on GLOF risk assessments at global scale, as presented by this study, are therefore definitely noteworthy.

SIGNIFICANCE OF THE WORK

As there is a large research community working on GLOFs and related topics, such a global study will be very significant for this field. The manuscript also presents the current state of scientific knowledge and open research gaps. It builds on (published and) publicly available, global data on glacial lakes and their historical evolution, population distribution and socio-economic indicators. These sources can be considered as established standards in this field. As mentioned before, past, current and future GLOF activity are topics that are addressed by many studies at different scales in recent years, but only few studies tackle these issues at global scale.

It is in the nature of global assessments that they need to rely on data that is globally available (often in a gridded format), and apply simple but robust approaches, that are efficient enough to be applied at global scale, but at the same time detailed enough to represent the relevant processes and their regional peculiarities based on the sparse data that is available. The present study transparently presents a methodology that relies on established datasets and aligns with other approaches. However, it has to commit some strong simplifications, that might compromise the significance of the results and conclusions, as detailed below.

DOES THE WORK SUPPORT THE CONCLUSIONS AND CLAIMS

Assessing the hazards of almost 15,000 glacial lakes requires some strong simplifications. Since assessing the likelihood of an outburst (=probably) and the distributed downstream intensity for each glacial lakes, simply lake number and lake area is taken from the lake inventory. The authors argue, that probability of an outburst is skipped, but intensity is anyway considered, since their approach focuses on "quantifying the intensity of an outburst should a GLOF occur" (L406-407). Of course downstream impacts are considered in their approach, since potential GLOF trajectories are intersected with population data, but this is part of the exposure (and vulnerability) element of risk. Intensity of hazard would be measured in flow height or maybe flow speed of the outburst flood, but this is not considered. This means, both factors of hazard (i.e. probability and intensity) are ignored. There are approaches published to get proxies for probability and intensity (cf. Furian et al., 2021 or Zheng et al. 2021), but none of these approaches has been applied globally. Also the 50km runout distance for all lakes is a very strong simplification. The authors base this on a study on GLOF risk in the Indian Himalayas (Dubey and Goyal, 2020), where this threshold of maximum reach is justified with the same three cases cited also here on L435 (actually as part of the exposure methodology, I suggest to move up to hazard). But in Fig. 2a, average GLOF runout distances for the Cordillera Blanca and HMA are indicated at much shorter distances (12 and 22km, respectively). For both the Andes and HMA, the largest bars representing exposed population in Fig 2a are actually beyond this average runout distances. A widely used alternative to determine maximum GLOF reach is minimum average slope angle between the source lake and the maximum reach, often chosen to be 2° or 3°. This might be an alternative here.

In my view, the chosen approach can be used for this study, but it needs to be reflected in the questions that are addressed and the conclusions that are drawn, especially regarding the simplifications related to the hazard element of risk: I argue, that due to the strong simplification of replacing GLOF hazard by the number and area of glacial lakes, the outcomes of the presented approach provide relevant information on exposure and vulnerability to GLOFs. But statements about GLOF risk are not very robust (also due to further methodological limitations, see below).

FLAWS IN THE DATA ANALYSIS, INTERPRETATION AND CONCLUSIONS

In my view, the way the normalization of hazard, exposure and vulnerability is done, should be revised. By applying Eq. 1 (L422), all values are normalized between 0 and 1. This leads to the strange effect, that a country (Uzbekistan) has a normalized hazard value of 0.000, despite the fact that it hosts glacial lakes. This is actually contradicting the above cited statement on L406-407, that the intensity of an outburst is considered should a GLOF occur: In that sense, if one (or several) glacial lakes exists in a catchment, hazard cannot be 0.000. Also at mountain range scale, GLOF hazard in the Alps is 1000 times lower (0.001) than in the PNW (1.00). It is very likely that GLOF hazard in the Alps indeed is lower than in the PNW, but the ratio is most likely heavily exaggerated.

I therefore suggest, at least for hazard, to apply a normalization comparing only the actual number (or area) to the maximum number (or area), i.e. ignoring the "Min" parameter in Eq. 1. This would reflect the fact, that the hazard is only 0 in case 0 lakes exist in a region/catchment.

Further I note that it is not declared if the hazard values in the results section and the discussion refer to hazard based on the number of lakes or the area of lakes (yN or yA, respectively, according to Eq. 1). Only supplementary figure 1 shows the difference between number of lakes and area of lakes, but it is unclear on which of the two all the results and discussions are based on.

The fact that all three elements of risk are normalized from 0-1, also has direct influence on the resulting risk. One of the main conclusions of the paper is, that "... the catchments most at-risk of GLOF impacts are not home to the largest, most numerous, or most rapidly expanding glacial lakes. Instead, we show it is the number and proximity of people to a glacial lake, and their capacity to cope with disaster that plays a central role in determining risk" (Abstract, L17-20, and similar in the conclusions on L199-201). This findings align with other, single-case GLOF risk evaluations (e.g. Huggel et al. 2020). But in the present study, risk is calculated as the multiplication of normalized hazard, exposure and vulnerability (Eq. 4, L497). So due to mathematical reasons it is obvious that hazard will only make one third of the final risk, and two thirds are determined by exposure (=the number of people) and vulnerability (their capacity to cope with disasters). Thus, although the conclusion is probably true, the result in this study might be mainly an effect of the applied methodology.

REPRODUCIBILITY

The methods are explained and described in a clear and transparent way, which makes the work reproduceable. Also, all the underlying data are accessible.

The only details which remains unclear to me are related to the assessment of vulnerability. First about the format: The CPI and HDI are values per country, that are then assigned to the population (per capita, so to say)? In the Data Availability section, it says "sub-national human development scores..." (L502). What means sub-national in this context? This should be made more explicit. Then the fact that (presumably) national averages are taken and applied to the mountain population (e.g. in Table S2) is subject to potentially strong biases, since the rural mountain population normally, or at least in many regions, has a lower HDI, higher illiterate population etc. than national averages. Actually, most of these parameters (safe drinking water, good sanitation, internet users etc.) might be substantially lower in the population that is potentially exposed to GLOFs than on national averages. This biases should at least be carefully discussed. In particular the "Urban population" factor makes no sense to consider if this are national averages. Argentina for

instance, just to pick an example, has according to Table S2 an urban population share of 92% (L531), this might be true for the entire country, but certainly not 92% of the population potentially exposed to GLOFs is living in urban areas (I assume). Further, I think Eq. 4 to calculate vulnerability needs to be adjusted. In its current form (L497), vulnerability is highest when HDI, CPI and SVI are low. This makes sense for HDI, but should be opposite for CPI (high corruption leads to high vulnerability). For SVI this is unclear, as it is unclear whether in Eq. 2 (L482) "-ve indicators" and "+ve indicators" refer to factors that reduce (red values in Table S2) or enhance (green values in Tab. S2) the capacity to cope with GLOFs.

RECOMMENDATIONS

As outlined in the beginning, the paper addresses a highly relevant topic. Assessing GLOF risk at global scale is a major contribution to the research community working in this and related fields. In particular I appreciate the recommendation on the future directions of work, addressing the need for more holistic approaches, evaluating risk based on all its component. This is definitely needed, in particular when deciding on priority regions for future work.

This given, but then also considering the concerns raised above, I suggest the following adjustment to be made:

- A slightly more sophisticated approach to estimate GLOF hazard, to go a bit beyond only relying on lake number/area for the hazard part. In case this is not feasible, I suggest to slightly adjust the scope (and title) of the paper and focus on the assessment of global GLOF exposure and vulnerability. Maybe this could be even extended to other floods. But the exposure and vulnerability aspects are really the novel and innovative part of the study in my view. Statements on GLOF risk are of course an even more interesting outcome, but based on the concerns raised above, I doubt that the current hazard assessment approach really allows to draw robust conclusions on GLOF risk.
- Adjust the normalization approach in a way that 0 really means zero hazard and zero exposure, respectively (i.e. no lake, or no potentially affected population).
- Clarify to which spatial resolution or entities (states, regions, sub-regions, gridded information) the parameters for the vulnerability assessment refer to, and carefully discuss potential biases that result from applying national averages to mountain population.
- Consider the detailed comments below, also regarding figures and tables.

I strongly encourage the authors to revise the manuscript accordingly, as this is a highly relevant study and important contribution to this research topic.

DETAILED COMMENTS

L15: Replace "future GLOFs" with "potential GLOFs" to avoid confusion. "Future GLOFs" could be misinterpreted as referring to potential GLOFs from future glacial lakes. Here and throughout the manuscript (also in L408, for instance).

L22: analyses (plural).

L27: Reference 11: In addition you might also want to cite the Nature paper by Hugonnet et al. (2021), which provides the latest numbers of glacier fluctuations at global scale.

L29-30: I suggest reformulating "... glaciers are retreating into deeper basal topography" and replace by something like "overdeepenings in the former glacier beds are uncovered in the course of glacier retreat" or similar.

L42: The number of 30,000 people has been revised, this is now considered as an overestimation. E.g. the death toll of the two Huascarán avalanches in 1961 and 1970 is expected to not have exceeded 7000 (Evans et al. 2009), which in older sources sums alone to 30,000 or more victims. Also the Palcacocha 1941 outburst, often numbered with several thousands of victims most likely didn't exceed 1800 victims (e.g. Mergili et al. 2019). I suggest to rephrase to "several thousands of victims".

L69: Give the full name of PNW when mentioning for the first time.

L84: Replace "Other" to a form that makes a full phrase.

L106: Insert a subtitle "Risk" (in the same format as Hazard, Exposure and Vulnerability

above).

L111-112: 2 million compared to 0.3 million would be "... more than six times the exposed population..." (instead of three times).

L136-138: The source cited here (Ref. 38) refers to an analysis of Web of Science until 2016. This is not necessarily reflecting the research hotspots of the last 5 years, which have probably seen a major increase in GLOF research items.

L162-164: Is it certain that mountain regions in Pakistan will also see an increase in population? In this study, exposure is measured via population density. Expansion of agriculture, development of new HEP sites, and growth of the tourism sector does not automatically increase population. Also, from the four cited references (28, 46-48), only the last one is on HMA, the others are general (47) or on the Andes (26 and 46). If this is referring not only to Pakistan but a general statement it should be made more clear. But then also tendencies of outmigration from mountain areas (trend for migration to cities) should be considered.

L174-176: I would not confirm that there is a lack of GLOF research in Pakistan. Maybe there are not too many classical GLOF hazard assessments, but there are the GLOF vulnerability projects implemented by UNDP, first a 4 million USD project funded by the Adaptation Fund (<https://www.adaptation-fund.org/project/reducing-risks-and-vulnerabilities-from-glacier-lake-outburst-floods-in-northern-pakistan/>), that was then scaled up by Green Climate fund to a 43 million USD project (<https://www.pk.undp.org/content/pakistan/en/home/projects/Glof-II.html>). This is probably the largest GLOF project worldwide.

L298-299: More bibliographic information needed for Ref. 35.

L406-407: "... and focus on quantifying the intensity of outburst should a GLOF occur but not the probability of the outburst occurring." This part of the sentence needs to be revised. The intensity of outburst should a GLOF occur is not assessed either, intensity in the context of hazard would mean quantitative (and spatially distributed) information on flow heights, flow speed, kinetic energy or similar physical parameters. Maybe revise to "quantifying the impacts or effects on the potentially affected population". But this is actually done under the exposure component of the methodology. In turn, the identification of the 50km downstream section, currently mentioned under the exposure part of methods (L432-435), could be moved here.

L480-482: It is unclear which of the SVI factors (the ones that reduce or enhance coping capacity) refer to -ve and +ve indicators, respectively, in Eq. 2. In the caption of Tab. S2 it says "Metrics in red increase vulnerability to GLOF, those in green reduce vulnerability to GLOF" (L532). To avoid confusion, I suggest to use the same formulation here (i.e. +ve and -ve increase or decrease vulnerability, rather than increase or decrease coping capacity, which is opposite).

L500-506: In the Data Availability section it is unclear where the factors of SVI are coming from. Is this included in UNDP's HDI datasets? Then this should be mentioned explicitly, otherwise the corresponding sources should be given. See also comments on spatial entities and resolution above (concerning "Sub-national human development scores, L503-504) where more details are needed either here or rather in the corresponding text further up.

FIGURES AND TABLES:

Besides the few details given below, the figures and table are clear and of sufficient quality and nicely illustrate the content of the manuscript.

Fig. 1a: I suggest to adjust the sizes of the different pies according to the number of total population exposed to GLOFs in each mountain range. This would make it more obvious where the exposure is highest and might even make the "global pie" in the lower left corner obsolete (especially since this pie again appears as Fig. 2b). Further I suggest to color the catchment with glacial lakes for "Other" mountain ranges in purple instead of grey in the map, as the corresponding pie and the legend.

Fig. 2a: Adjust the color for "Other" in the legend to purple. The light grey bars representing the total population (in the background) should not be wider than the other bars, this erroneously suggests a much larger population (due to the larger area of the

bar).

Fig. 2c can be skipped in my view, this is identical information to Fig. 1a.

Figure S1: On the y axis, change "Count" to "Count / Area".

In the table below it is again unclear if "Hazard score" is based on lake number or lake area. According to Eq. 1, one value for each should result. Further I don't understand why the lowest Hazard score (here for the Alps) isn't 0.000. If Eq. 1 is applied, the lowest value should always be 0.000 (and the highest 1.000, which is the case).

Figure S2: As in Fig. 2a, the background bars representing the total population per mountain range should not be wider than the other bars.

Figure S3: In the caption it should be mentioned why some countries are shaded in grey (I guess the ones that contain glacial lakes?).

Figure S4: It might be more illustrative and intuitive to plot the bars of the factors that increase GLOF vulnerability (i.e. the first four) as negative values.

Figure S5: Panel (d) is not needed in my view, this is all shown and included in (e). Why in (f), (g) and (h) only a subset of the above panel is shown? At least indicate/highlight in (e) the 5 sub-catchments that are picked for these three lowest panels.

Table S1: Please add a column with the catchment name, this would be extremely informative.

REFERENCES CITED IN THIS REVIEW

Dubey, S., & Goyal, M. K. (2020). Glacial Lake Outburst Flood Hazard, Downstream Impact, and Risk Over the Indian Himalayas. *Water Resources Research*, 56(4). <https://doi.org/10.1029/2019wr026533>

Evans, S. G., Bishop, N. F., Smoll, L. F., Murillo, P. V., Delaney, K. B., & Oliver-Smith, A. (2009). A re-examination of the mechanism and human impact of catastrophic mass flows originating on Nevado Huascarán, Cordillera Blanca, Peru in 1962 and 1970. *Engineering Geology*, 108(1–2), 96–118. <https://doi.org/10.1016/j.enggeo.2009.06.020>

Furian, W., Loibl, D., & Schneider, C. (2021). Future glacial lakes in High Mountain Asia: an inventory and assessment of hazard potential from surrounding slopes. *Journal of Glaciology*, 67(264), 653–670. <https://doi.org/10.1017/jog.2021.18>

Huggel, C., Carey, M., Emmer, A., Frey, H., Walker-Crawford, N., & Wallimann-Helmer, I. (2020). Anthropogenic climate change and glacier lake outburst flood risk: local and global drivers and responsibilities for the case of lake Palcacocha, Peru. *Natural Hazards and Earth System Sciences*, 20(8), 2175–2193. <https://doi.org/10.5194/nhess-20-2175-2020>

Hugonnet, R., McNabb, R., Berthier, E., Menounos, B., Nuth, C., Girod, L., et al. (2021). Accelerated global glacier mass loss in the early twenty-first century. *Nature*, 592(7856), 726–731. <https://doi.org/10.1038/s41586-021-03436-z>

Mergili, M., Pudasaini, S. P., Emmer, A., Fischer, J.-T., Cochachin, A., & Frey, H. (2019). Reconstruction of the 1941 GLOF process chain at Lake Palcacocha (Cordillera Blanca, Peru). *Hydrology and Earth System Sciences*, 24(1), 93–114. <https://doi.org/10.5194/hess-24-93-2020>

Zheng, G., Allen, S. K., Bao, A., Ballesteros-Cánovas, J. A., Huss, M., Zhang, G., et al. (2021). Increasing risk of glacial lake outburst floods from future Third Pole deglaciation. *Nature Climate Change*, 11(5), 411–417. <https://doi.org/10.1038/s41558-021-01028-3>

Reviewer #3 (Remarks to the Author):

- What are the noteworthy results?

The paper is a first global scale study on GLOF hazards and has come up with interesting analysis at catchment level to identify hotspots. The result shows GLOF risk is highest across the Andes and the authors state that there is a geographical disparity on research hotspots (Iceland, the North American Cordillera and HMA) and GLOF hotspots (Andes).

- Will the work be of significance to the field and related fields? How does it compare to the established literature? If the work is not original, please provide relevant references.

There are more rigorous assessments of potentially dangerous glacial lakes at regional level (e.g. Bajracharya, S.R. et. al (2020) Inventory of glacial lakes and identification of potentially dangerous glacial lakes in the Koshi, Gandaki, and Karnali River Basins of Nepal, the Tibet Autonomous Region of China, and India. Research Report. ICIMOD and UNDP). The topic is of significance to GLOF research community, however, the analysis is too generalized for any practical interventions on the basis of the findings.

- Does the work support the conclusions and claims, or is additional evidence needed?

The authors claim to have combined the most up-to-date hazard, exposure, and vulnerability data and ranked contemporary (2020) GLOF risk for glacial lake catchments at a global scale which will support in identifying higher priority zones for mitigation.

GLOF is a very complex phenomena which can be triggered by several factors and the impacts can vary depending upon different scenario (e.g. Sattar A et. al (2021) Modeling lake outburst and downstream hazard assessment of the Lower Barun Glacial Lake, Nepal Himalaya, Elsevier <https://doi.org/10.1016/j.jhydrol.2021.126208>). Rate of change in lake size and dam characteristics are important parameters for considering GLOF risks. The study uses only the raw number and area of glacial lakes per catchment to act as proxies for GLOF hazard which can be a serious limitation in the study results.

- Are there any flaws in the data analysis, interpretation and conclusions? - Do these prohibit publication or require revision?

The issues with the analysis carried out are:

Risk – Use of only the number and area of glacial lakes by catchment poses limitations as mentioned above.

Exposure – Adoption of 50 km reach and 1 km buffer of streams makes the analysis too coarse. However, the authors have provided their justification.

Vulnerability – It is seen that the countries with high Corruption Perception Index (CPI) are also the countries with low Human Development Index (HDI) and high Social Vulnerability Index (SVI). Using these correlated indices together could have issues of multicollinearity.

- Is the methodology sound? Does the work meet the expected standards in your field?

Mostly, GLOF risks have been studied using hydrodynamic models to identify the affected settlements and infrastructure. The methodology adopted in this study gives a very generalized picture of risks which can be very different in large scale analysis.

The indices are taken at the country level which could significantly differ from the actual situation in the buffer areas where the exposure population is calculated. This can create ecological fallacy related to erroneous conclusions emerging from cross-level inference, for example, when inferring results at local levels based on analysis at a more aggregate level.

**- Is there enough detail provided in the methods for the work to be reproduced?
The details are sufficient for reproducing the work.**

Reviewer #4 (Remarks to the Author):

This is a timely contribution to assess GLOF risk considering hazard, exposure and vulnerability at a global scale. It will facilitate to focus studies, monitoring networks and disaster risk reduction plans where they are most needed. Therefore, it will be useful for several disciplines including geosciences, engineering, and planning.

Here are some comments to hopefully correct some non-major issues:

1. Comments

As the authors correctly state in Suppl Info, channel gradient is key yet no consideration was given to estimate hazard with some consideration to it eg perhaps a combination of channel downstream distance with gradient. Given there are world DTMs available this could have been at least approximately done.

Secondly (and somewhat related to topography data), there should be also a clearer statement in the body of the submission on the probable overestimation of exposed population with the 1km buffer. Particularly for those catchments where as the authors mention in the Suppl Info the headwater channel due to steeper XS topography means less of a buffer is granted. It is understandable that the 1km resolution dataset doesn't allow more detail, but this shortcoming should be made explicit in the body of the manuscript not only in Supp Inf

L42-43: perhaps mention that a few large events are responsible for so many deaths in Cordillera Blanca?

2. Edits

L69: first appearance of PNW should be explicit it is Pacific-Northwest

L72, 110, 112, 113, 140: even if autocorrect changes Colombia to Columbia, this is something that at any level but particularly Nature, should occur, 5 times moreover. Let's be careful with names of countries, especially if it is signalled as one of the riskier ones. Please proofread better before submitting.

Fig3a: careful, figure shows shaded areas in Patagonian Andes where no glacial lakes exist (eg see Wilson et al), this might create confusion! Please correct.

Fig4a: National, not Global Population (upper rhs box)

RESPONSE TO REVIEWER COMMENTS

We would like to thank all 3 reviewers for their positive and constructive feedback. In particular, we thank Reviewer 2 for their detailed comments and suggestions on how to improve the manuscript and tighten its focus. Below, we provide a point-by-point response to each of the comments made by the reviewers, with our response in red. Line numbers refer to the manuscript version, where changes are highlighted in yellow.

In particular, we note that all reviewers, and particularly Reviewer 2, highlight concerns with the necessary and appropriate simplifications we make in assessing hazard to undertake a comparative global analysis, and the implications for this in terms of our focus on risk. As such, **in line with the suggestions of Reviewer 2, we have reframed this study to focus on the exposure and vulnerability of downstream populations to potential GLOFs.** To do this, we retain our generalised approach to assess hazard, but **reframe our questions to focus on the number of people downstream and their vulnerability to GLOFs.** making more qualified statements about 'risk' that note the lack of probability in our hazard analysis.

The Authors

Reviewer #2 (Remarks to the Author):

NOTEWORTHY RESULTS

The manuscript addresses the assessment of current risks posed by glacial lakes to downstream population at global scale. **Risk posed by glacial lakes is a hot topic** in earth- and geosciences, but also climate change impact research: Since several years, abundant studies are published on related topics at local to regional (continental scale) on glacial lakes and related hazards. **This is the first study, aiming at assessing the risk of glacial lakes outburst floods (GLOFs) at global scale,** and thereby not only focusing on the physical aspects of risk (i.e. hazard), but **considering also socio-economic drivers of risk,** by determining exposure and vulnerability of the population living downstream of such glacial lakes. Results on GLOF risk assessments at global scale, as presented by this study, are **therefore definitely noteworthy.**

R2.1. We would like to thank R2 for their positive overview and comments, and for highlighting that this is **the first study to consider socio-economic drivers of risk at a global-scale.** As R2 highlights previous work at the global scale has focussed almost exclusively on hazard, whilst those studies that do include socio-economic drivers of risk have only been at local-to-regional scale.

SIGNIFICANCE OF THE WORK

As there is a large research community working on GLOFs and related topics, **such a global study will be very significant for this field.** The manuscript also presents the current state of scientific knowledge and open research gaps. It builds on (published and) publicly available, global data on glacial lakes and their historical evolution, population distribution and socio-economic indicators. These sources can be considered as established standards in this field. As mentioned before, past, current and future GLOF activity are topics that are addressed by many studies at different scales in recent years, **but only few studies tackle these issues at global scale.**

It is in the **nature of global assessments that they need to rely on data that is globally available** (often in a gridded format), and **apply simple but robust approaches,** that are efficient enough to be applied at global scale, but at the same time detailed enough to represent the relevant processes and

their regional peculiarities based on the sparse data that is available. The present study **transparently presents a methodology that relies on established datasets and aligns with other approaches.** However, it has to commit some strong simplifications, that might compromise the significance of the results and conclusions, as detailed below.

R2.2. We would like to thank the reviewer for their **strongly positive comments on the significance and applicability of our work.** We agree that the global nature of this study requires strong simplifications and the combination of gridded, globally available data. We address each of the concerns below individually, but note here that the **comments by this reviewer have helped reframe and refocus this study** to one investigating the exposure and vulnerability of downstream communities to GLOF hazards, putting the **emphasis on global population and vulnerability data rather than GLOF hazard data.** Through this reframing, we intend to **limit the effect of the necessary simplifications we have had to make** in order to not compromise the significance of our results.

DOES THE WORK SUPPORT THE CONCLUSIONS AND CLAIMS

Assessing the hazards of almost 15,000 glacial lakes requires some strong simplifications. Since assessing the likelihood of an outburst (=probably) and the distributed downstream intensity for each glacial lakes, simply lake number and lake area is taken from the lake inventory. The authors argue, that probability of an outburst is skipped, but intensity is anyway considered, since their approach focusses on “quantifying the intensity of an outburst should a GLOF occur” (L406-407). Of course **downstream impacts are considered in their approach, since potential GLOF trajectories are intersected with population data,** but this is part of the exposure (and vulnerability) element of risk. Intensity of hazard would be measured in flow height or maybe flow speed of the outburst flood, but this is not considered. This means, both factors of hazard (i.e. probability and intensity) are ignored. There are approaches published to get proxies for probability and intensity (cf. Furian et al., 2021 or Zheng et al. 2021), but none of these approaches has been applied globally.

R2.1. We agree that assessing the hazard for such a large number of lakes over the entire globe requires strong simplifications. The **cited examples do provide methods to generate proxies for intensity and probability, however neither have been applied globally** and both themselves contain strong simplifications. For instance, Zheng et al. (2021) use simple area-volume relationships to convert mapped lake area to an estimated lake volume to represent outburst intensity. **We argue that applying a simple area-volume conversion consistently to all lakes globally would just scale our current hazard parameter, rather than offering additional data on intensity.** As such, our hazard results would not change by converting lake area to lake volume. In terms of probability of failure, here we consider it unknown for all and consequently are **focussing on what happens when a GLOF occurs,** rather than how likely that GLOF is. We now make this clear in the text through shifting our focus towards the exposure and vulnerability of downstream populations, as suggested by this reviewer. We note the work of both Zheng et al. (2021) and Furian et al (2021) who used simple proxies for landslides and ice avalanches into lakes but highlight this requires globally consistent DEMs which suffer from considerable artefacts in high mountain areas (Bolch and Loibl, 2018). **We add these references and detail this explanation in the text when describing our focus (Lines 71-76 and Lines 458-461).**

Also the 50km runout distance for all lakes is an very strong simplification. The authors base this on a study on GLOF risk in the Indian Himalayas (Dubey and Goyal, 2020), where this threshold of maximum reach is justified with the same three cases cited also here on L435 (actually as part of the exposure methodology, I suggest to move up to hazard

R2.2. We have moved this text into the hazard methods section (Lines 492-509).

But in Fig. 2a, average GLOF runout distances for the Cordillera Blanca and HMA are indicated at much shorter distances (12 and 22km, respectively). For both the Andes and HMA, the largest bars representing exposed population in Fig 2a are actually beyond this average runout distances. A widely used alternative to determine maximum GLOF reach is minimum average slope angle between the source lake and the maximum reach, often chosen to be 2° or 3°. This might be an alternative here.

R2.3. We appreciate these comments and suggestions. The 50 km runout chosen is a strong simplification but we argue that for an analysis such as ours, **some consistently applicable value needs to be defined.** We agree that a reach angle could be used as an alternative, but note that this would need to be applied consistently to all our input lakes. However, given the simplifications and assumptions required, **any changes in the number of exposed people as a result of moving from a fixed distance to a fixed reach angle are likely to be less than the uncertainty in the values themselves.**

We agree the 50 km runout is significantly larger than the past average distances noted in Fig 2, however, as now noted, this is a conservative estimate based on recently published works. **We believe it is prudent in this case to begin with a conservative distance that may over-predict than to miss areas/people at risk with a more liberal value.** We also note that the population distribution shown in Fig 2a shows consistent relative exposure over 5 km intervals within this range, highlighting that selecting a shorter distance would change the total number but not the relative distribution. **We also note that taking a larger distance than previous averages accounts for the increasing hazard from GLOFs due to increasing lake sizes,** which suggests that future GLOFs could be expected to have longer runout distances in future compared to previous events. We alter the text to make these points explicit as a justification for our choice (Lines 492-509).

In my view, the chosen approach can be used for this study, but it needs to be reflected in the questions that are addressed and the conclusions that are drawn, especially regarding the simplifications related to the hazard element of risk: I argue, that due to the strong simplification of replacing GLOF hazard by the number and area of glacial lakes, **the outcomes of the presented approach provide relevant information on exposure and vulnerability to GLOFs.** But statements about GLOF risk are not very robust (also due to further methodological limitations, see below).

R2.4. We appreciate and agree with the reviewer. From this comment, we retain our current approach to simplifying hazard, with the above described modifications to support our justification. **Based on this comment, we have reframed the manuscript as suggested, to focus explicitly on providing information on exposure and vulnerability to GLOFs** and we dampen and remove statements explicitly related to risk (Lines 16-21 and Lines 71-76). In doing so, we believe the reframed paper retains the novel and significant conclusions highlighted by all reviewers, whilst addressing the issues related to the simplification of the hazard analysis in terms of statements related to risk.

FLAWS IN THE DATA ANALYSIS, INTERPRETATION AND CONCLUSIONS

In my view, the way the normalization of hazard, exposure and vulnerability is done, should be revised. By applying Eq. 1 (L422), all values are normalized between 0 and 1. This leads to the strange effect, that a country (Uzbekistan) has a normalized hazard value of 0.000, despite the fact that it hosts glacial lakes. This is actually contradicting the above cited statement on L406-407, that the intensity of an outburst is considered should a GLOF occur: In that sense, if one (or several) glacial lakes exists in a catchment, hazard cannot be 0.000. Also at mountain range scale, GLOF hazard in the Alps is 1000 times lower (0.001) than in the PNW (1.00). It is very likely that GLOF

hazard in the Alps indeed is lower than in the PNW, but the ratio is most likely heavily exaggerated. I therefore suggest, at least for hazard, to apply a normalization comparing only the actual number (or area) to the maximum number (or area), i.e. ignoring the “Min” parameter in Eq. 1. This would reflect the fact, that the hazard is only 0 in case 0 lakes exist in a region/catchment.

R2.5. We agree that the suggested revision to normalising the data is a good approach and make the suggested changes to hazard and exposure (Line 485 and Line 526). We note that this normalisation approach was already applied to the vulnerability metric.

Further I note that it is not declared if the hazard values in the results section and the discussion refer to hazard based on the number of lakes or the area of lakes (y_N or y_A , respectively, according to Eq. 1). Only supplementary figure 1 shows the difference between number of lakes and area of lakes, but it is unclear on which of the two all the results and discussions are based on.

R2.6. Apologies that this is not clear, we have updated the text to make clear that we refer to hazard based on both the area (y_A) and number (y_N) of lakes and we use area as a proxy for flood intensity (Lines 489-491).

The fact that all three elements of risk are normalized from 0-1, also has direct influence on the resulting risk. One of the main conclusions of the paper is, that “... the catchments most at-risk of GLOF impacts are not home to the largest, most numerous, or most rapidly expanding glacial lakes. Instead, we show it is the number and proximity of people to a glacial lake, and their capacity to cope with disaster that plays a central role in determining risk” (Abstract, L17-20, and similar in the conclusions on L199-201). This findings align with other, single-case GLOF risk evaluations (e.g. Huggel et al. 2020). But in the present study, risk is calculated as the multiplication of normalized hazard, exposure and vulnerability (Eq. 4, L497). So due to mathematical reasons it is obvious that hazard will only make one third of the final risk, and two thirds are determined by exposure (=the number of people) and vulnerability (their capacity to cope with disasters). Thus, although the conclusion is probably true, the result in this study might be mainly an effect of the applied methodology.

R2.7. Whilst hazard does indeed only make up 1/3 of the total risk score for our study, we argue that this conclusion is still valid and corresponds with findings in other studies, as the reviewer notes. In particular, we highlight that the relative nature of our scores does allow locations with very high hazard but low exposure/vulnerability to retain high risk scores, for instance China has a higher risk score than Pakistan despite having significantly lower exposure and vulnerability (Lines 126-133). We also highlight the case of Greenland, which has the highest hazard scores but no exposure, thus resulting in a risk score of 0 (Lines 218-221). On the whole, our results show that in most cases, it is high(er) exposure scores that result in high(er) risk scores, which we believe justifies this statement.

REPRODUCIBILITY

The methods are explained and described in a clear and transparent way, which makes the work reproduceable. Also, all the underlying data are accessible.

The only details which remains unclear to me are related to the assessment of vulnerability. First about the format: The CPI and HDI are values per country, that are then assigned to the population (per capita, so to say)? In the Data Availability section, it says “sub-national human development scores...” (L502). What means sub-national in this context? This should be made more explicit.

R2.8. HDI scores are available for all nations at the first internal administrative sub-division unit. That is to say, the highest internal administrative level below national-scale used by

each country. For that reason, the spatial resolution varies by nation **provide a higher resolution than afforded by national level data**. We alter the text to make this clear (Lines 538-539).

Then the fact that (presumably) national averages are taken and applied to the mountain population (e.g. in Table S2) is subject to potentially strong biases, since the rural mountain population normally, or at least in many regions, has a lower HDI, higher illiterate population etc. than national averages. Actually, most of these parameters (safe drinking water, good sanitation, internet users etc.) might be substantially lower in the population that is potentially exposed to GLOFs than on national averages. This biases should at least be carefully discussed.

R2.9. We strongly agree that these measures are strong simplifications with bias. **To reduce this, we have used sub-national datasets, where possible, to provide greater resolution,** particularly in mountainous areas. We also now include a more detailed discussion of the potential limitations and biases within these data (Lines 565-572).

In particular the “Urban population” factor makes no sense to consider if this are national averages. Argentina for instance, just to pick an example, has according to Table S2 an urban population share of 92% (L531), this might be true for the entire country, but certainly not 92% of the population potentially exposed to GLOFs is living in urban areas (I assume).

R2.10. We agree that the vast majority of exposed populations are expected to be rural populations. **However, a measure of urban population fraction nationally has been shown to provide a reasonable assessment of a nation’s ability to respond to and cope with disasters such as GLOF** (Cutter et al., 2003). Those with higher proportions or urban population tend to have more centralised response mechanisms and access to greater resources to mobilise a response. This metric therefore represents the ability of the nation as a whole to cope with and respond to a GLOF, rather than a direct measure of the population sub-set that is directly exposed to GLOF. In this way, **our vulnerability measures is able to consider both the vulnerability of the directly impacted population and the capacity of the nation as a whole to respond appropriately**. This has been added at (Lines 568-572).

Cutter, S. L., Boruff, B. J., & Shirley, W. (2003). Social Vulnerability to Environmental Hazards. *Social Science Quarterly*, 84(2), 242-261.

Further, I think Eq. 4 to calculate vulnerability needs to be adjusted. In its current form (L497), vulnerability is highest when HDI, CPI and SVI are low. This makes sense for HDI, but should be opposite for CPI (high corruption leads to high vulnerability). For SVI this is unclear, as it is unclear whether in Eq. 2 (L482) “-ve indicators” and “+ve indicators” refer to factors that reduce (red values in Table S2) or enhance (green values in Tab. S2) the capacity to cope with GLOFs.

R2.11. We thank the reviewer for noting this error. Indeed, a high CPI does relate to a high vulnerability. **This is an error in the text but not in the data or application itself**. We have corrected the text accordingly (Line 581). We now also make clear in Eq.3 which indicators refer to increasing and decreasing capacity.

RECOMMENDATIONS

As outlined in the beginning, **the paper addresses a highly relevant topic. Assessing GLOF risk at global scale is a major contribution to the research community working in this and related fields**. In particular **I appreciate the recommendation on the future directions of work**, addressing the need for more holistic approaches, evaluating risk based on all its component. This is definitely

needed, in particular when deciding on priority regions for future work.

This given, but then also considering the concerns raised above, I suggest the following adjustment to be made:

- A slightly more sophisticated approach to estimate GLOF hazard, to go a bit beyond only relying on lake number/area for the hazard part. In case this is not feasible, **I suggest to slightly adjust the scope (and title) of the paper and focus on the assessment of global GLOF exposure and vulnerability.** Maybe this could be even extended to other floods. **But the exposure and vulnerability aspects are really the novel and innovative part of the study in my view.** Statements on GLOF risk are of course an even more interesting outcome, but based on the concerns raised above, I doubt that the current hazard assessment approach really allows to draw robust conclusions on GLOF risk.

R2.12. We thank the reviewer for their positive comments and helpful recommendations. In this case, we follow the recommendation of the reviewer to **refocus the paper and adjust the scope to focus on global exposure and vulnerability to GLOF.** We also alter the title to simply: “Glacial lake outburst floods threaten millions globally”. We appreciate the suggestion to extent to other floods, but believe this is out-of-scope of this study, noting that other recent works have been published in this space (McDermott, 2022; Tellman et al, 2021), and we include this references in the text. **We dampen statements related to risk** due to the hazard simplification, and now include recommendations on how to improve our simplistic hazard analysis to achieve more robust measures of risk.

McDermott, T.K.J. Global exposure to flood risk and poverty. Nat Commun 13, 3529 (2022). <https://doi.org/10.1038/s41467-022-30725-6>

- Adjust the normalization approach in a way that 0 really means zero hazard and zero exposure, respectively (i.e. no lake, or no potentially affected population).

R2.13. Done – see R2.7.

- Clarify to which spatial resolution or entities (states, regions, sub-regions, gridded information) the parameters for the vulnerability assessment refer to, and carefully discuss potential biases that result from applying national averages to mountain population.

R2.14. Done - see R2.10.

- Consider the detailed comments below, also regarding figures and tables.

R2.15. Done

I strongly encourage the authors to revise the manuscript accordingly, as this is a highly relevant study and important contribution to this research topic.

DETAILED COMMENTS

L15: Replace “future GLOFs” with “potential GLOFs” to avoid confusion. “Future GLOFs” could be misinterpreted as referring to potential GLOFs from future glacial lakes. Here and throughout the manuscript (also in L408, for instance).

R2.16. Done

L22: analyses (plural).

R2.17. Done

L27: Reference 11: In addition you might also want to cite the Nature paper by Hugonnet et al. (2021), which provides the latest numbers of glacier fluctuations at global scale.

R2.18. We thank the reviewer for this suggested inclusion – done

L29-30: I suggest reformulating "... glaciers are retreating into deeper basal topography" and replace by something like "overdeepenings in the former glacier beds are uncovered in the course of glacier retreat" or similar.

R2.19. Done

L42: The number of 30,000 people has been revised, this is now considered as an overestimation. E.g. the death toll of the two Huascarán avalanches in 1961 and 1970 is expected to not have exceeded 7000 (Evans et al. 2009), which in older sources sums alone to 30,000 or more victims. Also the Palcacocha 1941 outburst, often numbered with several thousands of victims most likely didn't exceed 1800 victims (e.g. Mergili et al. 2019). I suggest to rephrase to "several thousands of victims".

R2.20. Done, with references here included

L69: Give the full name of PNW when mentioning for the first time.

R2.21. Done

L84: Replace "Other" to a form that makes a full phrase.

R2.22. Done – we now refer to these as "High Arctic and Outlying Countries"

L106: Insert a subtitle "Risk" (in the same format as Hazard, Exposure and Vulnerability above).

R2.23. Done

L111-112: 2 million compared to 0.3 million would be "... more than six times the exposed population..." (instead of three times).

R2.24. Done – thank you for highlighting this error

L136-138: The source cited here (Ref. 38) refers to an analysis of Web of Science until 2016. This is not necessarily reflecting the research hotspots of the last 5 years, which have probably seen a major increase in GLOF research items.

R2.25. We now also include Emmer et al. (2022) which accounts for 2017-2021 and update the statistics accordingly.

L162-164: Is it certain that mountain regions in Pakistan will also see an increase in population? In this study, exposure is measured via population density. Expansion of agriculture, development of new HEP sites, and growth of the tourism sector does not automatically increase population. Also, from the four cited references (28, 46-48), only the last one is on HMA, the others are general (47) of on the Andes (26 and 46). If this is referring not only to Pakistan but a general statement it should be made more clear. But then also tendencies of outmigration from mountain areas (trend for migration to cities) should be considered.

R2.26. We appreciate the reviewers point here, but highlight that it is reasonably well established that mountain populations are moving into higher elevations as glaciers retreat, as the cited references show. Whilst these references do not explicitly refer to Pakistan, we believe it is fair to suggest that similar migrations will occur here as have been seen elsewhere. To clarify this, we add "as has been observed in other mountain regions globally." (Lines 199-200).

L174-176: I would not confirm that there is a lack of GLOF research in Pakistan. Maybe there are not too many classical GLOF hazard assessments, but there are the GLOF vulnerability projects

implemented by UNDP, first a 4 million USD project funded by the Adaptation Fund (<https://www.adaptation-fund.org/project/reducing-risks-and-vulnerabilities-from-glacier-lake-outburst-floods-in-northern-pakistan/>), that was then scaled up by Green Climate fund to a 43 million USD project (<https://www.pk.undp.org/content/pakistan/en/home/projects/Glof-II.html>). This is probably the largest GLOF project worldwide.

R2.27. This is a fair comment. We have revised this line to refer to limited *published* literature but note the recent large investments made in this space (Lines 211-213).

L298-299: More bibliographic information needed for Ref. 35.

R2.28. Done

L406-407: "... and focus on quantifying the intensity of outburst should a GLOF occur but not the probability of the outburst occurring." This part of the sentence needs to be revised. The intensity of outburst should a GLOF occur is not assessed either, intensity in the context of hazard would mean quantitative (and spatially distributed) information on flow heights, flow speed, kinetic energy or similar physical parameters. Maybe revise to "quantifying the impacts or effects on the potentially affected population". But this is actually done under the exposure component of the methodology. In turn, the identification of the 50km downstream section, currently mentioned under the exposure part of methods (L432-435), could be moved here.

R2.29. Done as suggested

L480-482: It is unclear which of the SVI factors (the ones that reduce or enhance coping capacity) refer to -ve and +ve indicators, respectively, in Eq. 2. In the caption of Tab. S2 it says "Metrics in red increase vulnerability to GLOF, those in green reduce vulnerability to GLOF" (L532). To avoid confusion, I suggest to use the same formulation here (i.e. +ve and -ve increase or decrease vulnerability, rather than increase or decrease coping capacity, which is opposite).

R2.30. Done – in Eq. 3 we now refer to enhancing and reducing indicators and the caption for Tab S2 has been updated

L500-506: In the Data Availability section it is unclear where the factors of SVI are coming from. Is this included in UNDP's HDI datasets? Then this should be mentioned explicitly, otherwise the corresponding sources should be given. See also comments on spatial entities and resolution above (concerning "Sub-national human development scores, L503-504) where more details are needed either here or rather in the corresponding text further up.

R2.31. Done

FIGURES AND TABLES:

Besides the few details given below, the figures and table are clear and of sufficient quality and nicely illustrate the content of the manuscript.

Fig. 1a: I suggest to adjust the sizes of the different pies according to the number of total population exposed to GLOFs in each mountain range. This would make it more obvious where the exposure is highest and might even make the "global pie" in the lower left corner obsolete (especially since this pie again appears as Fig. 2b). Further I suggest to color the catchment with glacial lakes for "Other" mountain ranges in purple instead of grey in the map, as the corresponding pie and the legend.

R2.32. We thank the reviewer for this helpful suggestion – done.

Fig. 2a: Adjust the color for “Other” in the legend to purple. The light grey bars representing the total population (in the background) should not be wider than the other bars, this erroneously suggests a much larger population (due to the larger area of the bar).

R2.33. Done

Fig. 2c can be skipped in my view, this is identical information to Fig. 1a.

R2.34. Removed

Figure S1: On the y axis, change “Count” to “Count / Area”.

R2.35. Done

In the table below it is again unclear if “Hazard score” is based on lake number or lake area.

R2.36. Clarified

According to Eq. 1, one value for each should result. Further I don't understand why the lowest Hazard score (here for the Alps) isn't 0.000. If Eq. 1 is applied, the lowest value should always be 0.000 (and the highest 1.000, which is the case).

R2.37. Updated normalisation method has now been applied as per this reviewers suggestion

Figure S2: As in Fig. 2a, the background bars representing the total population per mountain range should not be wider than the other bars.

R2.38. Done

Figure S3: In the caption it should be mentioned why some countries are shaded in grey (I guess the ones that contain glacial lakes?).

R2.39. We have expanded the figure caption to explain this.

Figure S4: It might be more illustrative and intuitive to plot the bars of the factors that increase GLOF vulnerability (i.e. the first four) as negative values.

R2.40. We appreciate this comment however, because the values are % total population for all indicators, we feel negative values are misleading. Instead, we now colour ‘negative’ bars as dashed and ‘positive’ bars as solid versions appropriate to their region to better distinguish.

Figure S5: Panel (d) is not needed in my view, this is all shown and included in (e).

R2.41. Done.

Why in (f), (g) and (h) only a subset of the above panel is shown?

R2.42. At the wider resolution, the areas being extracted are not visible.

At least indicate/highlight in (e) the 5 sub-catchments that are picked for these three lowest panels.

R2.43. Done. Explanatory text added to fig caption

Table S1: Please add a column with the catchment name, this would be extremely informative.

R2.44. We agree this would be useful however, very few of the 1089 catchments included have clearly identifiable names that we could use, and for some of the smaller catchments the names can vary even between local populations. We have therefore added catchment names where available and add a footnote to the table noting the absence of others.

REFERENCES CITED IN THIS REVIEW

We would like to thank the reviewer for highlighting these references, which we now include in our revised manuscript

Dubey, S., & Goyal, M. K. (2020). Glacial Lake Outburst Flood Hazard, Downstream Impact, and Risk Over the Indian Himalayas. *Water Resources Research*, 56(4).

<https://doi.org/10.1029/2019wr026533>

Evans, S. G., Bishop, N. F., Smoll, L. F., Murillo, P. V., Delaney, K. B., & Oliver-Smith, A. (2009). A re-examination of the mechanism and human impact of catastrophic mass flows originating on Nevado Huascarán, Cordillera Blanca, Peru in 1962 and 1970. *Engineering Geology*, 108(1–2), 96–118. <https://doi.org/10.1016/j.enggeo.2009.06.020>

Furian, W., Loibl, D., & Schneider, C. (2021). Future glacial lakes in High Mountain Asia: an inventory and assessment of hazard potential from surrounding slopes. *Journal of Glaciology*, 67(264), 653–670. <https://doi.org/10.1017/jog.2021.18>

Huggel, C., Carey, M., Emmer, A., Frey, H., Walker-Crawford, N., & Wallimann-Helmer, I. (2020). Anthropogenic climate change and glacier lake outburst flood risk: local and global drivers and responsibilities for the case of lake Palcacocha, Peru. *Natural Hazards and Earth System Sciences*, 20(8), 2175–2193. <https://doi.org/10.5194/nhess-20-2175-2020>

Hugonnet, R., McNabb, R., Berthier, E., Menounos, B., Nuth, C., Girod, L., et al. (2021). Accelerated global glacier mass loss in the early twenty-first century. *Nature*, 592(7856), 726–731.

<https://doi.org/10.1038/s41586-021-03436-z>

Mergili, M., Pudasaini, S. P., Emmer, A., Fischer, J.-T., Cochachin, A., & Frey, H. (2019). Reconstruction of the 1941 GLOF process chain at Lake Palcacocha (Cordillera Blanca, Peru). *Hydrology and Earth System Sciences*, 24(1), 93–114. <https://doi.org/10.5194/hess-24-93-2020>

Zheng, G., Allen, S. K., Bao, A., Ballesteros-Cánovas, J. A., Huss, M., Zhang, G., et al. (2021). Increasing risk of glacial lake outburst floods from future Third Pole deglaciation. *Nature Climate Change*, 11(5), 411–417. <https://doi.org/10.1038/s41558-021-01028-3>

Reviewer #3 (Remarks to the Author):

- What are the noteworthy results?

The paper is a first global scale study on GLOF hazards and has come up with interesting analysis at catchment level to identify hotspots. The result shows GLOF risk is highest across the Andes and the authors state that there is a geographical disparity on research hotspots (Iceland, the North American Cordillera and HMA) and GLOF hotspots (Andes).

R3.1. We would like to thank R3 for their positive overview and comments, and for highlighting that this is **the first study to consider socio-economic drivers of risk at a global-scale.**

- Will the work be of significance to the field and related fields? How does it compare to the established literature? If the work is not original, please provide relevant references.

There are more rigorous assessments of potentially dangerous glacial lakes at regional level (e.g. Bajracharya, S.R. et. al (2020) Inventory of glacial lakes and identification of potentially dangerous glacial lakes in the Koshi, Gandaki, and Karnali River Basins of Nepal, the Tibet Autonomous Region of China, and India. Research Report. ICIMOD and UNDP).

R3.2. We agree with the reviewer that more rigorous assessments do exist at the regional level. However, this is expected due to the ability to use higher resolution datasets and greater spatial heterogeneity. As R2 highlights, in order to apply our work at global scale a number of strong simplifications are needed and are **appropriate to provide a globally consistent study.**

The topic is of significance to GLOF research community. however, the analysis is too generalized for any practical interventions on the basis of the findings.

R3.3. We note R3 believes this work to be significant, however we disagree that the analysis does not allow practical interventions. As we note in our discussion, **a key outcome of this work is to highlight where more detailed and rigorous studies are urgently required,** and to highlight the disparity between where research is currently dominantly undertaken (HMA) and where risk is highest (Andes) (Lines 178-182). As such, a practical intervention from our work would be an increase in detailed local studies in the Andes and specifically the catchments we flag as being at high risk.

We believe, and both R2 and R4 agree, that taking a step back from local- and regional-scale studies and looking at a global-scale where we need to focus our limited resources is vital under increasing climate change, expanding lakes and increasing vulnerability. In particular, **we note that one of the highest risk catchments in our study is the Khyber Pakhtunkhwa** where recent (monsoon) flooding has caused considerable damage and loss of life. **More than US\$30M is currently being invested by the UN in GLOF projects in Pakistan and our work can help to identify other nations where similar investment is urgently needed.**

- Does the work support the conclusions and claims, or is additional evidence needed?

The authors claim to have combined the most up-to-date hazard, exposure, and vulnerability data and ranked contemporary (2020) GLOF risk for glacial lake catchments at a global scale which will support in identifying higher priority zones for mitigation.

GLOF is a very complex phenomena which can be triggered by several factors and the impacts can vary depending upon different scenario (e.g. Sattar A et. al (2021) Modeling lake outburst and downstream hazard assessment of the Lower Barun Glacial Lake, Nepal Himalaya, Elsevier <https://doi.org/10.1016/j.jhydrol.2021.126208>). Rate of change in lake size and dam characteristics are important parameters for considering GLOF risks. The study uses only the raw number and area of glacial lakes per catchment to act as proxies for GLOF hazard which can be a serious limitation in the study results.

R3.4. R2 raised similar concerns and we have addressed those concerns in our response to those comments above (response R2.3). In response to R3's comment about **rate of change being an important parameter, we strongly agree, and note that we explicitly evaluate this component in Fig 4** where we compare rate of change in lake area and number to rate of change in population by country.

- Are there any flaws in the data analysis, interpretation and conclusions? - Do these prohibit publication or require revision?

The issues with the analysis carried out are:

Risk – Use of only the number and area of glacial lakes by catchment poses limitations as mentioned above.

Exposure – Adoption of 50 km reach and 1 km buffer of streams makes the analysis too coarse. However, the authors have provided their justification.

R3.5. Both these concerns were also raised by R2 and we have addressed these comments above – response R2.3 addresses the use of number and area; response R2.5 addresses the 50 km runout distance.

Vulnerability – It is seen that the countries with high Corruption Perception Index (CPI) are also the countries with low Human Development Index (HDI) and high Social Vulnerability Index (SVI). Using these correlated indices together could have issues of multicollinearity.

R3.6. We have undertaken a collinearity analysis and include the results in the SI. We note that no significant collinearities were identified between these datasets. We explain this in **Lines 554-562**.

- Is the methodology sound? Does the work meet the expected standards in your field?

Mostly, GLOF risks have been studied using hydrodynamic models to identify the affected settlements and infrastructure. The methodology adopted in this study gives a very generalized picture of risks which can be very different in large scale analysis.

The indices are taken at the country level which could significantly differ from the actual situation in the buffer areas where the exposure population is calculated. This can create ecological fallacy related to erroneous conclusions emerging from cross-level inference, for example, when inferring results at local levels based on analysis at a more aggregate level.

R3.7. Similar concerns were raised in detail by R2 and we have provided detailed commentary on our response above in responses R2.10-R2.12.

- Is there enough detail provided in the methods for the work to be reproduced?

The details are sufficient for reproducing the work.

Reviewer #4 (Remarks to the Author):

This is a timely contribution to assess GLOF risk considering hazard, exposure and vulnerability at a global scale. It will facilitate to focus studies, monitoring networks and disaster risk reduction plans where they are most needed. Therefore, it will be **useful for several disciplines including geosciences, engineering, and planning.**

R4.1. We thank the reviewer for their positive comments here and their helpful recommendations below, and **note that R4 strongly agrees with R2 about the importance and significance of our work here at global-scale.**

Here are some comments to hopefully correct some non-major issues:

1. Comments

As the authors correctly state in Suppl Info, channel gradient is key yet no consideration was given to estimate hazard with some consideration to it eg perhaps a combination of channel downstream distance with gradient. Given there are world DTMs available this could have been at least approximately done.

R4.2. R2 highlights a similar concern. We address that concern above in response R2.5, but briefly here, the **available DEMs/DTMs at global level are both too coarse to provide an appropriate assessment and suffer from considerable artefacts in high mountain regions**, meaning such an analysis is likely to be fraught and potentially misleading. We state this in the manuscript at **Lines 458-461**.

Secondly (and somewhat related to topography data), there should be also a clearer statement in the body of the submission on the probable overestimation of exposed population with the 1km buffer. Particularly for those catchments where as the authors mention in the Suppl Info the headwater channel due to steeper XS topography means less of a buffer is granted. It is understandable that the 1km resolution dataset doesn't allow more detail, but this shortcoming should be made explicit in the body of the manuscript not only in Suppl Inf

R4.3. We believe we are clear in our Methods section on the relative strengths and weaknesses of this approach in terms of over- and under-estimating population exposure as a result of this buffer. We are however, now more explicit in the Introduction, Results and Discussions sections on the area from which exposure values have been calculated (**Lines 72-73 and Lines 89-90 and Line 192**).

L42-43: perhaps mention that a few large events are responsible for so many deaths in Cordillera Blanca?

R4.4. Done

2. Edits

L69: first appearance of PNW should be explicated it is Pacific-Northwest.

R4.5. Done

L72, 110, 112, 113, 140: even if autocorrect changes Colombia to Columbia, this is something that at any level but particularly Nature, should occur, 5 times moreover. Let's be careful with names of countries, especially if it is signalled as one of the riskier ones. Please proofread better before submitting.

R4.6. Our apologies, we have corrected this throughout the text.

Fig3a: careful, figure shows shaded areas in Patagonian Andes where no glacial lakes exist (eg see Wilson et al), this might create confusion! Please correct.

R4.7. We are confused by this comment since Wilson et al (2018) report substantial numbers of glacial lakes in the Central and Patagonian Andes, reporting that glacial lakes in the region increased in number by 43% between 1986 and 2016.

Wilson et al (2018) Glacial lakes of the Central and Patagonian Andes, *Global and Planetary Change*, 162, p.275-291.

Fig4a: National, not Global Population (upper rhs box)

R4.8. Corrected

REVIEWER COMMENTS

Reviewer #2 (Remarks to the Author):

GENERAL

The authors provide a revised version of the original manuscript, addressing the comments raised in the first review. They provide detailed responses to all comments in the rebuttal letter with references to the changes implemented in the manuscript.

The comments raised in the first round of reviews are taken up positively and constructively, and I appreciate that the authors submitted a revised version while agreeing to most of my suggestions and reasonably justifying the few other cases, where they prefer to stick to their original version. In that sense, the authors agree that the applied hazard determination approach is too simple to fulfill the definition of a hazard and risk assessment, respectively, and to shift the focus on the exposure and vulnerability components of GLOF risk.

The revised version is now clearly improved, however, in the submitted manuscript, the announced shift from risk to mainly the exposure and vulnerability component is not yet implemented sufficiently. Still, the text is formulated in a way that a full risk assessment has been done, including a hazard, exposure and vulnerability component. While the exposure and vulnerability component are now ready to be published, the analyses under the hazard component still do not qualify for a full risk analysis (as the authors agree in their rebuttal letter).

While the analyses and figures mostly can remain as they are, the text still needs revisions to reflect adequately what kind of conclusions can be drawn where the limits of the analysis are. I am aware that my comments might be a bit pedantic, but risk and hazard (as a risk component) are clearly defined terms in this context and the applied approach does not meet the requirements of these definitions (mainly related to the hazard component, and thus the full risk assessment).

The easiest way to solve this is to use alternative terminologies - without necessarily cutting on the relevance of the related results. In that sense, I suggest to use "lake conditions" or "lake predisposition" instead of "hazard", and "potential GLOF impact" or "GLOF danger" or "GLOF threat" instead of "GLOF risk" in the context of this study. Further, the combination of "exposure" and "vulnerability" could be called "damage potential".

Before I give some more detailed suggestion for reformulations, I want to touch on the potential to assess GLOF hazard even at global scale. In my original review this is addressed in the first paragraph of DOES THE WORK SUPPORT THE CONCLUSIONS AND CLAIMS, and is responded by the authors in R2.1 (the second R2.1, there are two responses with that number): The authors are absolutely correct when they state that in the context of this study, there is no difference between using lake area or lake volume (as Zheng et al. (2021) did) as a proxy for potential GLOF intensity. However, the reason, why I mentioned the approaches of Zheng et al. (2021) and Furian et al. (2021) is not because they use lake volume instead of area, but because they both present approaches for assessing the potential of mass movement impacts (rock/ice avalanches and landslides) into the glacial lakes based on topographical analyses of the lake surroundings, as also noted further down by the authors. Such approaches can be used as a proxy for lake outburst probability, which is lacking in the present manuscript. It is true that these approaches require globally available DEMs, but this is not per se a limitation for even a global application. Required DEMs do not necessarily need to be globally highly consistent and there is a large number of global studies in glaciated environments available, which are based on freely available DEM data at global scale. The mentioned approaches are not very sensitive to local artefacts, as they are mostly based on spatially aggregated parameters for a buffer zone or the contributing catchment or similar, further the output would be a normalized value. Nevertheless, I understand that it is not feasible to include such an analysis in the present study and I don't request this. But since it isn't done (for good reasons), the related outcomes should not be called hazard and risk.

SUGGESTED CHANGES IN THE TEXT

Considering the above, I suggest the following changes in the abstract:

L14-15: "... contemporary exposure and vulnerability to GLOFs has never been quantified".

L21: "... are most exposed and vulnerable to potential GLOF impacts" or "... have the highest GLOF damage potential".

L24: "... which are key factors for determining the level of potential GLOF impact. By ranking GLOF threat ...".

L27: "... with some catchments with highest potential GLOF impact globally".

In the main text, rewording and reformulations in this sense are needed to replace the terms hazard or risk, respectively. This also applies to section titles (L78, L122, and in the Methods section L452 and L583). Changes are also needed in the following lines:

L68 (maybe "up-to-date lake data"), **L75, L76** (maybe "to capture potential GLOF impacts through time").

The paragraph on L79-86 should be reformulated accordingly. Further, **L123, L128, L129, L132, L133, L135, L136, L141, L142, L156/157** (title and caption of Fig. 3), **L176, L167, L168, L172, L173, L180, L181, L182, L185-187** (caption of Fig. 4 "...exposed to" instead of "at risk of" on L186), **L192, L211, L217, L220** (twice), **L222, L228** (here is suggest to skip "..., and thus the risk"), **L238, L246, L249-251** (several instances), and **L452** (title) should be revised accordingly.

In the section on L452-469, the difference of what is analyzed here (lake predisposition, or however it is going to be called eventually) to "hazard" according to its definition, should be elaborated.

L583-583: Reformulate to "GLOF threat", "GLOF danger", "GLOF damage potential" or "potential GLOF impact", respectively.

FIGURES and TABLES

Titles and captions of Fig. 3, figure SF3, figure SF6, and supplementary Table 1 should be revised accordingly, to avoid the term "GLOF risk".

Other than that, the figures have been revised and adjusted adequately. The only detail I still believe is not 100% correct, is the width of the grey "global" bars in Figs. 1b, 2, SF1 and SF2, as mentioned already in the first review: Although the width of these bars has been slightly adjusted, the grey bars (mostly representing the "global" numbers) should not be wider than the individual colored bars, representing regions or countries. Using wider bars leads to much larger areas of the bars and thus to a visual confusion: Such wide bars are suggesting to represent a much larger number (of lakes, people, etc.), than it actually is the case. My suggestion is therefore, to use a grey bar next to the colored bars, width the same width as the colored bars (instead of a wider grey bar in the background).

I am still convinced that this study will be of high relevance and attendance, once it will be published with the suggested adjustments.

REFERENCES CITED IN THIS REVIEW

Furian, W., Loibl, D., & Schneider, C. (2021). Future glacial lakes in High Mountain Asia: an inventory and assessment of hazard potential from surrounding slopes. *Journal of Glaciology*, 67(264), 653–670. <https://doi.org/10.1017/jog.2021.18>

Zheng, G., Allen, S. K., Bao, A., Ballesteros-Cánovas, J. A., Huss, M., Zhang, G., et al. (2021). Increasing risk of glacial lake outburst floods from future Third Pole deglaciation. *Nature Climate Change*, 11(5), 411–417. <https://doi.org/10.1038/s41558-021-01028-3>

Reviewer #3 (Remarks to the Author):

I would like to thank the authors for addressing all the comments positively. Considering the limited availability of consistent data at global scale, the paper gives a good overview of global scenario of potential GLOF risk. As mentioned by the authors, the results can be useful to prioritize areas for future research.

Reviewer #4 (Remarks to the Author):

Thank you, the manuscript has been greatly improved, and I stand by my first review (with apologies on unintended spelling slips!) on the value to publish this paper. However based on my experience and what this journal stands on, I still believe there are a couple of areas where it needs further details before it can be published. I will follow as much as I can my review comments and related the Response codes used by the authors below.

1. Possible use of channel gradient: R4.2 (and R2.3 not 2.5) in my view correctly assess that the global DTMs have artefacts in steep mountain regions. Particularly, canyons where the usual 90m (even 30m GDEM) resolution has problems to include the channel (in more than one lateral pixel). We can only hope we can have soon a higher resolution global DTM that will allow better analyses, and perhaps this can be further stressed in the recommendations.

2. Coarseness of 1km buffer: with R4.3 and more explicit edits mentioned therein now help the reader to consider the probable estimation error produced by this coarse buffer (for example, consider canyons mentioned in previous point). Thank you.

3. Fig3 shaded Patagonia: perhaps my comment should have been more explanatory for those unfamiliar to (Chilean) Patagonia. In Fig 3, the shaded areas include large coastal areas without glaciers (& glacial lakes) in Chilean Patagonia regions of Aysen & Magallanes. Wilson et al (2018) that I cited partially and you kindly cited more completely, helps to make a better judgement of this, but checking any decent map of Chile or Chilean glaciers (eg Barcaza et al 2017) should be checked by the authors. Please correct, though I am afraid this connects to the next.

Barcaza, G., Nussbaumer, S. U., Tapia, G., Valdés, J., García, J-L., Videla, Y., Albornoz, A. and Arias, V. (2017): Glacier inventory and recent glacier variations in the Andes of Chile, South America. *Ann. Glaciol.*, 58, 166-180, doi:10.1017/aog.2017.28.

4. Catchments: unfortunately this submission is based on Yan et al 2019, which really is not very good relative to identifying catchments correctly. In fact, as this paper & database implies starting from its title, it provides vector layer info for Water Resources Zones divisions, NOT catchments. This is part of the reason of the error referred in comment 3 immediately above with Chilean Patagonian regions of Aysen & Magallanes. But also the errors in the table just above Supplementary Table S1: many on the 50 higher risk "catchments" list are NOT really catchments. Just to give examples of Chile and Italy where this reviewer has more experience:

Risk Rank vs "Catchment Name*"

6 CHL 51203030111 (39117km² from shp) encompasses most of southern Magallanes!

22 ITA 40208060200 (6075km²) covers Olona/Lambro in Po basin though not exactly

23 CHL 51202050211 (53429km²) covers most of NW Magallanes!

29 CHL 51301020211 (34677 km²) is most of the whole region of Aysen!

49 ITA 40208140000 (2476km²) appears to include Sangone/Chisola/Lemina

This reviewer strongly advises to revise this thoroughly, and provide a better presentation (strengths, weaknesses, biases and approximations) of the data and results in this matter.

Yan, D. et al. A data set of global river networks and corresponding water resources zones divisions. *Sci. Data* 6, 1–11 (2019).

In sum, in this reviewer's opinion, the above comments 3 & 4 need to be fully addressed for this submission to be publishable.

RESPONSE TO REVIEWER COMMENTS ROUND 2

We would like to thank all 3 reviewers for their continued positive and constructive feedback. In particular, we thank Reviewers 2 and 4 for their detailed comments and suggestions. Below, we provide a point-by-point response to each of the comments made by the reviewers, with our response in red numbered by reviewer and comment. Line numbers refer to the manuscript version. Note that in the manuscript where we have altered and replace figures, we have retained only the new version of the figure for clarity.

The Authors

Reviewer #2 (Remarks to the Author):

GENERAL

The authors provide a revised version of the original manuscript, **addressing the comments raised in the first review.** They provide detailed responses to all comments in the rebuttal letter with references to the changes implemented in the manuscript.

The **comments raised in the first round of reviews are taken up positively and constructively,** and I appreciate that the authors submitted a revised version while **agreeing to most of my suggestions and reasonably justifying the few other cases,** where they prefer to stick to their original version. In that sense, the authors agree that the applied hazard determination approach is too simple to fulfill the definition of a hazard and risk assessment, respectively, and to shift the focus on the exposure and vulnerability components of GLOF risk.

The revised version is now clearly improved, however, in the submitted manuscript, the announced shift from risk to mainly the exposure and vulnerability component is not yet implemented sufficiently. Still, the text is formulated in a way that a full risk assessment has been done, including a hazard, exposure and vulnerability component. While **the exposure and vulnerability component are now ready to be published,** the analyses under the hazard component still do not qualify for a full risk analysis (as the authors agree in their rebuttal letter).

While **the analyses and figures mostly can remain as they are,** the text still needs revisions to reflect adequately what kind of conclusions can be drawn where the limits of the analysis are. I am aware that my comments might be a bit pedantic, but risk and hazard (as a risk component) are clearly defined terms in this context and the applied approach does not meet the requirements of these definitions (mainly related to the hazard component, and thus the full risk assessment).

R2.1. We would like to thank R2 again for the strongly positive review and for their helpful suggestions and comments. As in the previous round of revisions, we have accepted most of the suggested ammendments and have provided a more detailed response to each point below.

The easiest way to solve this is to use alternative terminologies - **without necessarily cutting on the relevance of the related results.** In that sense, I suggest to use “lake conditions” or “lake predisposition” instead of “hazard”, and “potential GLOF impact” or “GLOF danger” or “GLOF threat” instead of “GLOF risk” in the context of this study. Further, the combination of “exposure” and “vulnerability” could be called “damage potential”.

R2.2. We appreciate this simple, helpful suggestion and have made the suggested changes, using GLOF lake conditions and GLOF danger throughout in place of hazard and risk, respectively. We also clearly highlight these changes and the reasoning in the Methods (Lines 486-489 and Lines 612-617). We do retain occasional mention of hazard and risk when referring to previously published

studies or to potential future avenues for research, where full hazard and risk could be determined, particularly for more local scale studies.

Before I give some more detailed suggestion for reformulations, I want to touch on the potential to assess GLOF hazard even at global scale. In my original review this is addressed in the first paragraph of DOES THE WORK SUPPORT THE CONCLUSIONS AND CLAIMS, and is responded by the authors in R2.1 (the second R2.1, there are two responses with that number): **The authors are absolutely correct when they state that in the context of this study, there is no difference between using lake area or lake volume** (as Zheng et al. (2021) did) as a proxy for potential GLOF intensity. However, the reason, why I mentioned the approaches of Zheng et al. (2021) and Furian et al. (2021) is not because they use lake volume instead of area, but because they both present approaches for assessing the potential of mass movement impacts (rock/ice avalanches and landslides) into the glacial lakes based on topographical analyses of the lake surroundings, as also noted further down by the authors. Such approaches can be used as a proxy for lake outburst probability, which is lacking in the present manuscript. It is true that these approaches require globally available DEMs, but this is not per se a limitation for even a global application. Required DEMs do not necessarily need to be globally highly consistent and there is a large number of global studies in glaciated environments available, which are based on freely available DEM data at global scale. The mentioned approaches are not very sensitive to local artefacts, as they are mostly based on spatially aggregated parameters for a buffer zone or the contributing catchment or similar, further the output would be a normalized value. **Nevertheless, I understand that it is not feasible to include such an analysis in the present study and I don't request this.** But since it isn't done (for good reasons), the related outcomes should not be called hazard and risk.

R2.3. We thank the reviewer for this detailed explanation. This highlights an important point that is a valuable avenue for future work that could enable our current work to extend fully into the risk space by accounting for probability.

SUGGESTED CHANGES IN THE TEXT

Considering the above, I suggest the following changes in the abstract:

L14-15: "... contemporary exposure and vulnerability to GLOFs has never been quantified".

R2.4. Done

L21: "... are most exposed and vulnerable to potential GLOF impacts" or "... have the highest GLOF damage potential".

R2.5. Done

L24: "... which are key factors for determining the level of potential GLOF impact. By ranking GLOF threat ...".

R2.6. Done – we use danger as opposed to threat (see R2.2).

L27: "... with some catchments with highest potential GLOF impact globally".

R2.7. Done

In the main text, rewording and reformulations in this sense are needed to replace the terms hazard or risk, respectively. This also applies to section titles (L78, L122, and in the Methods section L452 and L583).

R2.8. Done

Changes are also needed in the following lines:

L68 (maybe "up-to-date lake data"), L75, L76 (maybe "to capture potential GLOF impacts through time").

R2.9. Done

The paragraph on L79-86 should be reformulated accordingly.

R2.10. Done – we change replace hazard with lake conditions throughout.

Further, L123, L128, L129, L132, L133, L135, L136, L141, L142, L156/157 (title and caption of Fig. 3), L176, L167, L168, L172, L173, L180, L181, L182, L185-187 (caption of Fig. 4 “...exposed to” instead of “at risk of” on L186), L192, L211, L217, L220 (twice), L222, L228 (here is suggest to skip “..., and thus the risk”), L238, L246, L249-251 (several instances), and L452 (title) should be revised accordingly.

R2.11. All done – and we thank R2 for their thorough identification of these points of text!

In the section on L452-469, the difference of what is analyzed here (lake predisposition, or however it is going to be called eventually) to “hazard” according to its definition, should be elaborated.

R2.12. We have reformulated the text on lines 486-488 to highlight that we use GLOF lake conditions throughout due to the lack of probability in our scoring method.

L583-583: Reformulate to “GLOF threat”, “GLOF danger”, “GLOF damage potential” or “potential GLOF impact”, respectively.

R2.13. Done – we specifically describe why we prefer GLOF danger to GLOF risk

FIGURES and TABLES

Titles and captions of Fig. 3, figure SF3, figure SF6, and supplementary Table 1 should be revised accordingly, to avoid the term “GLOF risk”.

R2.14. Done

Other than that, **the figures have been revised and adjusted adequately**. The only detail I still believe is not 100% correct, is the width of the grey “global” bars in Figs. 1b, 2, SF1 and SF2, as mentioned already in the first review: Although the width of these bars has been slightly adjusted, the grey bars (mostly representing the “global” numbers) should not be wider than the individual colored bars, representing regions or countries. Using wider bars leads to much larger areas of the bars and thus to a visual confusion: Such wide bars are suggesting to represent a much larger number (of lakes, people, etc.), than it actually is the case. My suggestion is therefore, to use a grey bar next to the colored bars, width the same width as the colored bars (instead of a wider grey bar in the background).

R2.15. Done

I am still convinced that this study will be of high relevance and attendance, once it will be published with the suggested adjustments.

REFERENCES CITED IN THIS REVIEW

Furian, W., Loibl, D., & Schneider, C. (2021). Future glacial lakes in High Mountain Asia: an inventory and assessment of hazard potential from surrounding slopes. *Journal of Glaciology*, 67(264), 653–670.

<https://doi.org/10.1017/jog.2021.18>

Zheng, G., Allen, S. K., Bao, A., Ballesteros-Cánovas, J. A., Huss, M., Zhang, G., et al. (2021). Increasing risk of glacial lake outburst floods from future Third Pole deglaciation. *Nature Climate Change*, 11(5), 411–417. <https://doi.org/10.1038/s41558-021-01028-3>

Reviewer #3 (Remarks to the Author):

I would like to thank the authors for addressing all the comments positively. Considering the limited availability of consistent data at global scale, the paper gives a good overview of global scenario of potential GLOF risk. As mentioned by the authors, **the results can be useful to prioritize areas for future research.**

R3.1. We thank R3 for their positive comments on the revised version of this manuscript based on their early helpful suggestions

Reviewer #4 (Remarks to the Author):

Thank you, the manuscript has been greatly improved, and I stand by my first review (with apologies on unintended spelling slips!) on the value to publish this paper. However based on my experience and what this journal stands on, I still believe there are a couple of areas where it needs further details before it can be published. I will follow as much as I can my review comments and related the Response codes used by the authors below.

R4.1. We thank R4 for their positive comments on the revised version of this manuscript based and their early helpful suggestions which helped to greatly improve the manuscript.

1. Possible use of channel gradient: R4.2 (and R2.3 not 2.5) in my view **correctly assess that the global DTMs have artefacts in steep mountain regions.** Particularly, canyons where the usual 90m (even 30m GDEM) resolution has problems to include the channel (in more than one lateral pixel). We can only hope we can have soon a higher resolution global DTM that will allow better analyses, and perhaps this can be further stressed in the recommendations.

R4.2. We agree with R4 on the issue of artefacts within global DTMs in steep mountain regions, and thus the infeasible inclusion of this into our global study, which also agrees with R2, above.

2. Coarseness of 1km buffer: with R4.3 and more explicit edits mentioned therein **now help the reader to consider the probable estimation error produced by this coarse buffer** (for example, consider canyons mentioned in previous point). Thank you.

R4.3. We thank R4 for their previous suggestion, which helped improve this portion of the text.

3. Fig3 shaded Patagonia: perhaps my comment should have been more explanatory for those unfamiliar to (Chilean) Patagonia. In Fig 3, the shaded areas include large coastal areas without glaciers (& glacial lakes) in Chilean Patagonia regions of Aysen & Magallanes. Wilson et al (2018) that I cited partially and you kindly cited more completely, helps to make a better judgement of this, but checking any decent map of Chile or Chilean glaciers (eg Barcaza et al 2017) should be checked by the authors. Please correct, though I am afraid this connects to the next.

Barcaza, G., Nussbaumer, S. U., Tapia, G., Valdés, J., García, J-L., Videla, Y., Alborno, A. and Arias, V. (2017): Glacier inventory and recent glacier variations in the Andes of Chile, South America. Ann. Glaciol., 58, 166-180, doi:10.1017/aog.2017.28.

R4.4. We appreciate R4s helpful comments and expertise in identifying this effect with the use of Water Resource Zones. We agree that in the named regions, the shapefiles do result in large unglaciated regions being included. Nevertheless, we highlight that our use of a 50 km cut off distance negates this issue by ensuring that only the population within 50 km of a glacial lake are included in our calculations (lines 534-538). We choose to colour the entire Water Resource Zone in Figs 1 and 3 for visual clarity only, given the global and regional scale of these figures. We now make this clear in figure 3 caption (lines 168-170). For example, Catchment ITA 40208060200 (highlighted by R4 in the comment below) does cover most of Olona/Lambro, however only population upstream of ~

Lago di Como and Lago Maggiore are considered in the exposure calculation, without including any portions of Milan city or its outer suburbs. Similarly, ARG 51301060200 (Santa Cruz 'catchment' in Argentina) is shown reaching to the Atlantic Ocean, but only population upstream of ~Tres Lagos are included in the calculation. We do now more clearly describe the Water Resource Zones and their weaknesses in our methods section (detailed below in R4.5).

4. Catchments: unfortunately this submission is based on Yan et al 2019, which really is not very good relative to identifying catchments correctly. In fact, as this paper & database implies starting from its title, it provides vector layer info for Water Resources Zones divisions, NOT catchments. This is part of the reason of the error referred in comment 3 immediately above with Chilean Patagonian regions of Aysen & Magallanes. But also the errors in the table just above Supplementary Table S1: many on the 50 higher risk "catchments" list are NOT really catchments. Just to give examples of Chile and Italy where this reviewer has more experience:

Risk Rank vs "Catchment Name*"

6 CHL 51203030111 (39117km² from shp) encompasses most of southern Magallanes!

22 ITA 40208060200 (6075km²) covers Olona/Lambro in Po basin though not exactly

23 CHL 51202050211 (53429km²) covers most of NW Magallanes!

29 CHL 51301020211 (34677 km²) is most of the whole region of Aysen!

49 ITA 40208140000 (2476km²) appears to include Sangone/Chisola/Lemina

This reviewer strongly advises to revise this thoroughly, and provide a better presentation (strengths, weaknesses, biases and approximations) of the data and results in this matter.

Yan, D. et al. A data set of global river networks and corresponding water resources zones divisions. *Sci. Data* 6, 1–11 (2019).

R4.5. We agree that the data of Yan et al (2019) are limited, since they do not really represent accurate 'catchments' per se. In that regard, we have altered our use of 'catchment(s)' to 'basin(s)' in the appropriate locations throughout to avoid confusion with true river catchments. We also make clear in our methods section the limitations of using Water Resource Zones rather than catchments (Lines 497-503 and Lines 534-538). We refer back to R4.4 for a more detailed response as to the effect of this on our results.

In sum, in this reviewer's opinion, the above comments 3 & 4 need to be fully addressed for this submission to be publishable.

R4.6. We thank R4 for highlight these 2 comments as being particularly important. We have now addressed those in detail, as above.